# How far away are truly hyperparameter-free learning algorithms?

**Priya Kasimbeg**  *kasimbeg@google.com*
*Google DeepMind*

**Vincent Roulet**  *vroulet@google.com*
*Google DeepMind*

**Naman Agarwal\***

**Sourabh Medapati**  *smedapati@google.com*
*Google DeepMind*

**Fabian Pedregosa**  *pedregosa@google.com*
*Google DeepMind*

**Atish Agarwala**  *thetish@google.com*
*Google DeepMind*

**George E. Dahl**  *gdahl@google.com*
*Google DeepMind*

**Reviewed on OpenReview:** *https://openreview.net/forum?id=6BlOCx5c5T*

## Abstract

Despite major advances in methodology, hyperparameter tuning remains a crucial (and expensive) part of the development of machine learning systems. Even ignoring architectural choices, deep neural networks have a large number of optimization and regularization hyperparameters that need to be tuned carefully *per workload* in order to obtain the best results. In a perfect world, training algorithms would not require workload-specific hyperparameter tuning, but would instead have default settings that performed well across many workloads. Recently, there has been a growing literature on optimization methods which attempt to reduce the number of hyperparameters—particularly the learning rate and its accompanying schedule. Given these developments, how far away is the dream of neural network training algorithms that completely obviate the need for painful tuning?

In this paper, we evaluate the potential of learning-rate-free methods as components of hyperparameter-free methods. We freeze their (non-learning rate) hyperparameters to default values, and score their performance using the recently-proposed ALGOPERF: Training Algorithms benchmark. We found that literature-supplied default settings performed poorly on the benchmark, so we performed a search for hyperparameter configurations that performed well across all workloads simultaneously. The best "ALGOPERF-calibrated" learning-rate-free methods had much improved performance but still lagged slightly behind a similarly calibrated NADAMW baseline in overall benchmark score. Our results suggest that there is still much room for improvement for learning-rate-free methods, and that testing against a strong, workload-agnostic baseline is important to improve hyperparameter reduction techniques.

---

*Work done while at Google DeepMind

# 1 Introduction

Achieving good results with deep learning requires finding reasonable settings for the large number of hyperparameters[1] that control the model configuration and training pipeline. Finding good hyperparameter settings is far from easy, so researchers who are unable to tune well enough often end up with worse results and more difficult scientific comparisons. Unfortunately, as models get larger and more costly to train, tuning becomes even more difficult because running large numbers of experiments is no longer feasible.

The predominant approach to hyperparameter selection in deep learning is a mixture of idiosyncratic human judgment and semi-automated search methods. Unfortunately, relying on too much subjective judgment both inhibits scientific reproducibility and risks achieving poor results, since there is no reason to think typical researchers should be particularly adept at selecting good hyperparameter values. This is a problem both for solving individual problems but also for scientific progress at large. The difficulty of accounting for hyperparameter search when comparing methods has likely contributed to many contradictions, false conclusions, and dead ends in the machine learning literature.

In practice, researchers often use some (potentially iterative) combination of the following strategies: (1) tuning by hand and selecting points to try based on their own intuition, (2) tuning using low-dimensional sweeps over a few hyperparameters, selecting both which ones to tune and their search ranges based on human judgment, or (3) tuning using blackbox search algorithms, including Bayesian optimization. Although Bayesian optimization (Mockus et al., 1978) and other similar techniques (e.g. Li et al. (2018)) can be effective when computational resources are abundant, they provide no guidance on how to construct the search spaces they require as input (Ariafar et al., 2022). Furthermore, when computational resources are limited, the success of Bayesian optimization methods strongly depend on the quality of the input search space. This issue is exacerbated in popular, more simplistic alternatives such as grid search and (quasi-)random search (Bergstra and Bengio, 2012; Bousquet et al., 2017). In effect, all these methods convert the problem of finding good hyperparameter settings into the problem of finding a good search space—which must once again be drawn from the mysterious well of expert intuition.

There are two routes to reducing the amount of human judgment needed in hyperparameter search: the first is to develop more automatic and efficient tuning methods, and the second is to reduce the number of hyperparameters. Although ultimately these approaches are two sides of the same coin, eliminating hyperparameters emphasizes pushing the tuning procedure inside the training algorithm, and is the path we are concerned with in this work. Even if completely eliminating all training algorithm hyperparameters isn't feasible, partial progress would be worthwhile. For example, discovering *which* hyperparameters are most essential to tune can lead to more efficient and robust methods, and help us start hyperparameter search from better points.

The real pain point of using training algorithms that require careful tuning is not the need for hyperparameter tuning *per se*, but the need for *workload-specific* tuning. For example, if a training algorithm required extensive tuning, but we could perform that tuning *once* on a small problem and reuse the hyperparameter settings successfully on radically different workloads, then the algorithm might as well be hyperparameter-free. Then we could tune, say, a model trained to classify MNIST digits (LeCun et al., 1998), and find good configurations of training algorithm hyperparameters for, say, training large scale machine translation models. The issue is not so much the number of training algorithm hyperparameters that need to be tuned as how much their optimal settings vary across workloads. The recent excitement in the literature surrounding scaling heuristics (Kaplan et al., 2020; Hoffmann et al., 2022; Wortsman et al., 2023) is fundamentally related to cross-workload hyperparameter transfer. There the goal is to use small models to optimize configurations for larger, more costly models while keeping the data distribution and basic architectural choices the same.

Within the deep learning literature, there is a growing body of work that attempts to eliminate optimization hyperparameters—most commonly, the base learning rate. The learning rate is perhaps the single most important training algorithm hyperparameter in most popular training algorithms (Bengio, 2012). Several

---

[1] As is common in the deep learning literature, we use the term "hyperparameter" to mean any configuration parameter of the training pipeline (e.g. learning rate). Our usage deviates from the original definition of "hyperparameter" as a parameter of a prior distribution in Bayesian machine learning.

heuristics have been proposed to adjust the learning rate during training according to a schedule (Loshchilov and Hutter, 2016; Smith, 2017; Li and Arora, 2019) that is a function of some desired number of training iterations, or training horizon, that is assumed to be known in advance. Such schedules typically exhibit two monotonic phases: a warm-up phase where the learning rate slowly increases up to some peak value, and a decay phase where the learning rate decreases to a small value, generally (near) zero (e.g. the schedule used in Goyal et al. (2017)). However, even these methods still require practitioners to identify an overall learning rate scale — the base learning rate — which must be tuned per workload.

The premise of learning-rate-free methods is to remove the need for a base learning rate (and ideally, the accompanying schedule) via changes to the training algorithm. Successful establishment of such methods would remove a leading contributor to hyperparameter tuning toil, and go a long way towards making tuning easier in practice. Removing the learning rate and other hyperparameters could enable better "default" optimization and regularization settings, which work well across many workloads.

In this paper, we study how useful recent learning-rate-free optimizers (Section 2.2) are as a step towards the more ambitious goal of removing workload-specific hyperparameter tuning from effective training algorithms. The goal of this work is to empirically study these learning-free-learning algorithms. Providing insights into the training dynamics that determine their performance is beyond the scope of this work. We take the position that we must study a diverse set of workloads to shed light on how effective these approaches are at reducing the need for hyperparameter search. The AlgoPerf: Training Algorithms benchmark (Dahl et al., 2023) and, in particular, its track for self-tuning/hyperparameter-free algorithms provides a perfect test bed to put learning-rate-free optimizers through their paces. We study the cross-workload performance of single hyperparameter settings and show the following:

- Learning-rate-free methods using default hyperparameter values from the literature perform poorly compared to AdamW/NadamW baselines that use a single set of evidence-based defaults (fixed across all workloads).

- Some learning-rate-free methods greatly benefit from *calibration* of regularization parameters, while others have no settings which generalize well. A simple search reveals configurations which are better across all workloads simultaneously.

- The best learning-rate-free methods using these new AlgoPerf-calibrated configurations are competitive with (but not clearly better than) similarly calibrated AdamW and NadamW methods.

In particular, we find that achieving good performance on AlgoPerf tasks still requires a choice equivalent to a training horizon or learning rate decay schedule, which are workload-dependent. Our results highlight the importance of benchmarking algorithms on a diverse set of workloads using well-posed tuning rules. The current crop of learning-rate-free methods is a promising step towards hyperparameter free methods, but there is still much work to be done to beat a natural baseline algorithm. Our work also suggests that tuning algorithms simultaneously across many workloads can reveal more useful default settings than those provided by papers or codebases.

## 2 What are hyperparameter-free methods?

For the purposes of this work, we assume that training algorithms operate on a fully specified *workload*—a combination of model architecture, loss function, evaluation metric, and dataset. We assume the purpose of a *training algorithm* is to produce a set of model parameters which performs well on the validation set in accordance with the evaluation metric. This choice of framing implies that workload-agnostic regularization methods, such as weight decay and dropout, are part of the training algorithm, but architectural design choices such as the number of layers or where to place batch normalization (Ioffe, 2015) layers are part of the workload definition (as are application-specific methods for improving generalization performance, such as most forms of data augmentation). The best training algorithms may not be the best *optimization algorithms*, as minimizing the training loss is only a means to an end.

Within this setting, we define a *hyperparameter-free*, or *self-tuning* training algorithm as one which works well across a variety of workloads without requiring *any* workload-specific choices to be made by the user. In principle, such algorithms could be developed using hyperparameter search on a small set of workloads, and would then need to generalize to a much broader set of workloads without any further tuning. In our work, we will use the same set of workloads to discover and test hyperparameter-free methods (to be discussed more concretely in Section 3).

One question raised by our operational definition of a hyperparameter-free training algorithm is how to handle the termination of training. Is the number of training steps a choice made by the user? In our view, a user is free to run a training algorithm for as long as they wish, but the duration of training, or training horizon, becomes a meaningful hyperparameter of the training algorithm only when the training algorithm isn't an "interruptible" or "anytime" algorithm. Unfortunately, most practical algorithms use an optimizer with some sort of learning rate schedule which decreases to 0 at the end of training—defining a *training horizon*. Such algorithms can achieve quite different results when trained with different horizons, with optimal training often occuring at some intermediate horizon value. We will return to this point in Section 2.2, and will include information about a "good" training horizon for standard methods as part of our workload specification.

## 2.1 Hyperparameter-free training algorithms from workload-agnostic settings

Although by our definition there are no truly hyperparameter-free training algorithms in wide use in deep learning today, there *are* methods that might be useful in building hyperparameter-free training algorithms, namely learning-rate-free optimizers. Learning-rate-free methods remove the base learning rate, which already provides some hyperparameter reduction; they also use information from the specific workload instance and current loss landscape to set the learning rate, which suggests a potential for automatic workload-specific adaptation.

A learning-rate-free method (or indeed, any optimizer) becomes a hyperparameter-free training algorithm by fixing the remaining optimizer parameters across all workloads. We can then ask the following questions:

- Can learning-rate-free methods perform well across many workloads without workload-specific hyperparameter tuning?

- Do learning-rate-free methods with fixed hyperparameters outperform standard methods with fixed hyperparameters across a diverse set of workloads?

Answering these questions quantitatively on a diverse set of benchmark workloads can give us insight into whether or not learning-rate-free optimizers are bringing us closer to hyperparameter-free training algorithms.

## 2.2 Candidate algorithms

**Adaptive Algorithms: the Common Baseline**   The difficulty of training deep networks with a simple stochastic gradient descent has led to the development of adaptive methods, the most popular of which is Adam (Kingma and Ba, 2014). Adaptivity here amounts to preconditioning the given stochastic gradients by an estimate of their second uncentered moment. In practice, Adam and its main variants have been the main optimizers used in deep learning on a broad range of tasks, with small variations (Schmidt et al., 2021). In our work we focus on two variants: Adam with decoupled weight decay (ADAMW, (Loshchilov and Hutter, 2017)) and Adam with Nesterov momentum (NADAMW, (Dozat, 2016)). Adam tunes the direction of the update in a geometry-dependent way, but still requires a base learning rate to set the overall *scale* of the update—a separate problem tackled by the learning-rate-free methods we detail below.

**Tuning Learning Rate by Estimating Initial Distance to Minimizer.**   A first line of learning-rate-free methods, also known as stepsize tuners (DoG, DoWG, D-ADAPT, PRODIGY presented below) are built on top of a theoretical analysis of a gradient descent on a convex non-smooth function. Consider a function $f$ optimized by a series of parameter iterates $w_k$ of the form $w_{k+1} = w_k - \eta_k g_k$ where $g_k$ is a (sub)gradient of $f$ at $w_k$. If $f$ is convex and $G$-Lipschitz-continuous, the optimal constant learning rate for a fixed horizon $K$

is then $\eta_k = D/(G\sqrt{K})$, where $D = \|w_0 - w_\star\|_2$ is the Euclidean distance between the initial point and a minimizer $w_\star$ of $f$ (Nesterov, 2018). Adaptive gradient methods such as AdaGrad adapt the learning rate to the unknown quantity $G$ with the help of accumulated gradients.

A more recent line of work takes a step further and estimates the distance $D$ from the initial point $w_0$ to any solution $w_\star$. *Distance over Gradients (*DoG*)* (Ivgi et al., 2023) considered simply estimating $D$ by the maximum distance between the iterates and the initial point. *Distance over Weighted Gradients (*DoWG*)* (Khaled et al., 2023) proposed to refine the adaptive rule used by AdaGrad equipped with DoG to take into account the varying learning rates. *D-Adaptation (*D-ADAPT*)* (Defazio and Mishchenko, 2023) exploits the convexity and Lipschitz-continuity of a theoretical objective to build a lower bound on $D$. PRODIGY (Mishchenko and Defazio, 2023) considers improving on D-ADAPT (Defazio and Mishchenko, 2023) by correcting the estimates of second moments in adaptive methods. We studied all of these algorithms in our initial experiments in Section 4; as we will see, PRODIGY is the most promising in this family due to its compatibility with preconditioners like ADAM and is the focus of experiments in Section 5.

**Continuous Coin Betting.** Continuous Coin Betting (CoCoB) (Orabona and Tommasi, 2017) builds upon a gambling viewpoint of stochastic optimization to adapt the learning rate. At each step, and for each coordinate, the amount of progress along the subgradient direction is seen as a bet whose scale depends on past and current subgradient values and their estimated scales. The optimal betting strategy is derived theoretically and adapted to the problem at hand by online estimates of the coordinate-wise gradient scales. Contrary to other algorithms like PRODIGY, D-ADAPT, MoMo or MECHANIC, COCOB does not use the same base scaling as Adam and rather defines its own adaptivity recipe to per-coordinate gradients.

**Mechanic.** The MECHANIC learning rate tuner (Cutkosky et al., 2023) (Algo. 3) is also designed from a study of theoretical convergence rates in a convex optimization framework. Its setup departs slightly from the aforementioned line of work by considering directly an online convex optimization framework as previously done in Adagrad (Duchi et al., 2011). Most importantly, MECHANIC is designed as a *wrapper* around any base optimizer, enabling a seamless integration of additional heuristics such as warmups or annealing schedules. The tuner is designed to guarantee that the produced iterates are, theoretically, not worse than the base optimizer.

**Model-Based Momentum.** The MoMo (Model-based Momentum) optimizer (Schaipp et al., 2023) (Algo. 4) defines iterates by minimizing a model of the loss function function based on an *a priori* lower bound of the loss and past loss values and gradients. For problems where the optimal loss value is non-zero, MoMo allows for online estimation of a tighter lower bound. In the absence of additional momentum or preconditioning techniques, MoMo reduces to *Polyak stepsizes* (Polyak, 1964), a celebrated technique to set stepsizes from convex optimization that has been adapted to stochastic optimization in deep learning recently (Berrada et al., 2020; Loizou et al., 2021).

### 2.3 Learning-rate-free algorithm implementation details

Despite being pitched as learning-rate-free, D-ADAPT, PRODIGY, MECHANIC, and MoMo[2] still recommend using a predetermined schedule, with a warmup and/or a decay phase. We will refer to such schedules as *relative learning rate schedules* to distinguish them from traditional, absolute learning rate schedules. The most common form this schedule takes is a real number, computed as a function of the number of steps, that is multiplied into the update provided by the optimizer, just before these updates are applied to the parameters. In other words, learning-rate-free methods remove the *base* learning rate which sets the overall scale, but not the changes in learning rate over training —including the training horizon over which decay takes the learning rate to 0. These schedules are themselves not exempt from tuning; we will return to this point in Section 5.

---

[2]For MoMo (Schaipp et al., 2023), the authors argue that the algorithm naturally tends to implement a schedule for their experiments on feedforward networks. However, they also use an explicitly applied schedule for their experiments on transformers and diffusion models.

Table 1: Main additional hyperparameters of learning-rate-free methods.

| Algorithm | Base Learning Rate | Initial Estimates |
|---|---|---|
| DoG | Schedule-based, peak lr 1. | Square distance to minimizer, $10^{-6}(1 + \|w_0\|)/\sqrt{\|g_0\|^2 + 10^{-8}}$ |
| DoWG | Schedule-based, peak lr 1. | Square distance to minimizer, $10^{-4}$ |
| Prodigy/D-Adapt | Schedule-based, peak lr 1. | Distance to minimizer, $10^{-6}$ |
| Mechanic | Schedule-based, peak lr 1. | Initial scaling, $10^{-4}$ |
| COCOB | Constant/schedule-based lr 1. | Gradient scales, 1 |
| MoMo | Constant/schedule-based lr 1e-2/1. | Lower bound on objective, 0 |

Additionally, all the candidate algorithms work by estimating some unknown global quantity of the problem, such as the distance between the initial point and the minimizer. These quantities are eventually estimated using data but require an initialized value. We used values from the literature; the parameters and their values are described in Table 1.

Mechanic also has a decay parameter $\lambda_{\mathrm{mech}}$ which is used internally by the algorithm, and an integer $k_{\mathrm{mech}}$ defining the number of estimates used to approximate the scaling factor. Both of which have been kept to their default values ($\lambda_{\mathrm{mech}} = 10^{-2}$ and $k_{\mathrm{mech}} = 6$ respectively). In practice, Prodigy actually also requires an additional modification to incorporate a warmup schedule (called `safeguard_warmup` in the official implementation), otherwise the algorithm easily diverges. COCOB uses an additional hyperparameter $\alpha = 100$ that defines the fraction of the current estimate of the gradient magnitude to cap the denominator or the scaling of the updates, see (Orabona and Tommasi, 2017, Section 6).

All algorithms presented above leave aside the tuning of (i) any *weight decay*, (ii) the *momentum or exponential moving average parameter of the gradients* (used in SGD with momentum or Adam), and (iii) the *exponential moving average parameter of the second moment* estimate in Adam. We will explore the consquences of tuning these additional parameters in Section 5.

## 3 Methods

### 3.1 Workloads and training

We studied the performance of the candidate algorithms by training 8 workloads from the MLCommons® AlgoPerf: Training Algorithms benchmark on the self-tuning track (Dahl et al., 2023). The AlgoPerf workloads span a diverse collection of architectures and datasets across image, text, speech, and graph domains. Each workload is specified by a Dataset, Model, Loss, and evaluation metric (see Table 2 for a summary). The workloads in this study were trained on TPUv2 2x2 slices (16GB HBM per chip), with the exception of Criteo 1TB DLRM small which was trained on TPUv3 4x4 (32GB HBM per chip). This is in contrast with the original benchmark in which algorithms are trained on 8x Nvidia V100 GPUs.

The interface for the training algorithms is designed to be as workload-agnostic as possible. At each training step the interface exposes the loss function, loss value, gradients, gradient function, model weights, and learning rate. Training algorithms also have access to one additional piece of information: the workload-dependent *maximum steps*. We set this quantity by first starting with the per-workload, maximum *wall-clock time* set by the AlgoPerf self-tuning ruleset (Dahl et al., 2023). We converted this time measurement to a steps measurement by measuring the number of steps AdamW took in the allotted time on the AlgoPerf competition hardware. This measurement gave us an upper bound on the number of steps that an algorithm should take to reach the target on the workload. We note that the algorithms we studied all took about 10% more time per step than AdamW; therefore the maximum step restriction is generous compared to the equivalent time restriction.

The maximum number of steps plays a key role in the training algorithms: we used it to define workload-specific training horizons for algorithms with relative schedules. Since in our study we used a single set of hyperparameters across workloads, we defined learning rate schedules as follows. The learning rate $\eta_t$ at a

step $t$ is given by $\eta_t = f(t/t_{\text{hor}}) \cdot \eta_{\text{alg},t}$, where $f : [0,1] \rightarrow [0,1]$ gives the shape of the schedule, $t_{\text{hor}}$ is the training horizon, and $\eta_{\text{alg},t}$ is the base learning rate computed each step by the algorithm. The training horizons themselves are defined relative to the maximum steps; that is, for a given workload $i$, $t_{\text{hor}} = \alpha_i t_{\text{max},i}$ where $t_{\text{max},i}$ is the maximum steps for that workload, and $\alpha_i$ is some fraction from 0 to 1. In our experiments we fixed $\alpha_i$ across workloads to avoid an additional per-workload tuning burden.

**Fixed hyperparameter training.** In order to explore the potential of current training algorithms as *hyperparameter-free* methods, we conducted experiments where all hyperparameters were *fixed across workloads*. Our experiments give a sense of how these algorithms generalize across a variety of workloads without tuning. We specified the optimizer's hyperparameters, regularization hyperparameters, and a learning rate schedule (or equivalent), and applied those same settings across all workloads individually. The only workload-specific data were the measurements returned each step by the training interface, as well as the horizon of the learning rate schedule which was defined as a fraction of the maximum steps per workload. Our experiments were inspired by the ALGOPERF competition's self-tuning track, which similarly scores training algorithms on their ability to generalize across workloads without workload-specific hyperparameter tuning.

Table 2: **Summary of the workloads in the AlgoPerf benchmark**. The possible losses are the cross-entropy loss (CE), the mean absolute error (L1), and the Connectionist Temporal Classification loss (CTC). The evaluation metrics additionally include the structural similarity index measure (SSIM), the error rate (ER), the word error rate (WER), the mean average precision (mAP), and the bilingual evaluation understudy score (BLEU). This table also lists the batch sizes we used for training each of the workloads.

| Task | Dataset | Model | Loss | Metric | Target Metric | Max Steps | Batch Size |
|---|---|---|---|---|---|---|---|
| Clickthrough rate prediction | CRITEO 1TB | DLRMSMALL | CE | CE | 0.1236 | 31,998 | 262,144 |
| MRI reconstruction | FASTMRI | U-NET | L1 | SSIM | 0.724 | 108,567 | 32 |
| Image classification | IMAGENET | RESNET-50 | CE | ER | 0.226 | 559,998 | 1024 |
| | | VIT | CE | ER | 0.227 | 559,998 | 1024 |
| Speech recognition | LIBRISPEECH | CONFORMER | CTC | WER | 0.086 | 240,000 | 256 |
| | | DEEPSPEECH | CTC | WER | 0.120 | 144,000 | 256 |
| Molecular property prediction | OGBG | GNN | CE | mAP | 0.281 | 240,000 | 512 |
| Translation | WMT | TRANSFORMER | CE | BLEU | 30.85 | 399,999 | 128 |

## 3.2 Evidence based calibration of learning-rate-free methods

One way to frame the search for a good hyperparameter-free method derived from a training algorithm with multiple hyperparameters is as a search for a *calibrated default*—a single hyperparameter configuration that shows good results on a diverse set of benchmark workloads. We conducted a series of experiments to find such hyperparameter settings for our candidate algorithms. Our setup doesn't have any held-out workloads; we used the whole set of ALGOPERF base workloads to find our cross-workload hyperparameter settings. This is a necessary first step towards solving the general problem of using one set of workloads to find evidence based calibrated defaults which also work well on previously untested workloads.

We performed hyperparameter search on a ADAMW and NADAMW baselines, as well as one learning-rate-free algorithm from each candidate family. While the learning-rate-free methods automatically set the base learning rate, finding good values for the remaining hyperparameters that govern the optimizer, regularization, warmup fraction, and training horizon is a nontrivial task.

To find good hyperparameters, we used a search procedure to maximize combined performance across the ALGOPERF tasks. At the end of this process we obtain ALGOPERF-*calibrated* hyperparameters for each learning-rate-free method, which represent our best-effort hyperparameter-free algorithms. The search procedure consisted of the following steps, described in detail below:

1. Set training horizon: specify $\alpha$ and calculate $t_{\text{hor}}$, which determines the number of steps of the cosine decay schedule.

2. Quasi-random search: given a fixed training horizon, run hyperparameter configurations sampled at random from a search space.

3. Hyperparameter selection: given a set of hyperparameter configuration trial results, select the top-3 configurations based on aggregate performance on all workloads.

**Setting Training Horizons.** The inclusion of training horizon information is perhaps the biggest difference between our experimental setup and the situation which practitioners face "in the wild." A number of the algorithms that we studied use a learning rate decay schedule, which requires us to define a *training horizon*—the number of steps before the learning rate decays to 0. The success of these algorithms is highly sensitive to the choice of training horizon. We chose the training horizon as fractions based on the maximum runtimes provided by ALGOPERF. We converted the ALGOPERF maximum runtimes to a maximum number of steps (as detailed in Section 3.1), which was then used to pick training horizons per workload. We explored performance over a fixed set of horizons corresponding to 33%, 50% and 66% of the maximum steps (per-workload). Preliminary experiments suggested that increasing the training horizon beyond 66% does not result in any gains in time-to-target metrics, so we did not further analyze performance on longer training horizons.

**Random Search.** For each algorithm, we first perform a quasi-random search (Bousquet et al., 2017) on each of the training horizons specified. To perform the random search we devised a search space over the hyperparameters. This space contains both discrete and continuous sets depending on the hyperparameter. The search space is defined in Table 3; base learning rate is only tuned for the ADAMW/NADAMW baselines. Note that the Label Smoothing hyperparameter is only relevant to the workloads that support that regularization method in the current implementation of the benchmark workloads, namely IMAGENET RESNET-50, IMAGENET VIT, OGBG GNN, WMT TRANSFORMER. From this search space we drew 200 points at random, where each point contains the hyperparameter settings to configure a single set of experiments on the 8 ALGOPERF workloads. We performed this search on all of the algorithms at least on the 50% training horizon. For algorithms that yielded promising results on the 50% training horizon we also performed this search on the 33% and 66% horizons.

Table 3: Search space for evidence-based search procedure.

| Parameter | Scale | Range |
|---|---|---|
| Base Learning Rate | Log | [1e-4, 5e-2] |
| Warmup | Discrete | {0.02, 0.05, 0.1} |
| Weight Decay | Log | [1e-5, 0.5] |
| $1 - \beta_1$ | Log | [1e-3, 1.0] |
| $1 - \beta_2$ | Log | [1e-3, 1.0] |
| Dropouts (tied) | Discrete | {0.0, 0.1} |
| Label Smoothing | Discrete | {0.0, 0.2} |

**Hyperparameter Selection.**

In line with Anonymous (2024), we used a cost function based on the geometric mean of a time-to-target metric to rank the 200 points. We then took the top 3 configurations under this metric and computed their ALGOPERF benchmark scores. This required re-training with 5 independent random seeds. For each algorithm-training horizon pair, the configuration with the highest benchmark score was chosen for the final comparisons. See Section 3.3 for more details on the two scoring metrics.

**Baseline selection.** Many of the candidate learning-rate-free algorithms that we consider in this work can be interpreted as wrappers around some base optimizer, which is often ADAMW in the default implementation. Therefore it is natural to compare algorithms against ADAM baselines. We compared against both ADAMW and NADAMW, as there is evidence in recent literature that NADAMW may outperform ADAMW (Dahl

et al., 2023). To configure the ADAMW and NADAMW baseline we used the evidence-based-search procedure detailed above, including the base learning rate as a search dimension. Note that we used the same number of search points, spread over one more dimension of search. The top 3 search candidates for each method (and the eventual baselines chosen) are detailed in Table 7.

### 3.3 Measuring performance

Given a set of promising algorithms, there are many methods of measuring their performance. In both our hyperparameter search as well as our final analyses we used a series of performance measurements that build on each other:

**Time to target.** We set a validation evaluation metric target value, and measured how quickly the target is reached (or if it is reached at all within the budget). This metric is the basis for ALGOPERF scoring, and is only useful when a large enough fraction of algorithms train successfully (i.e. achieve the validation metric goal within the time limit) a large enough fraction of the time. In that setting, this metric measure training speed and how often training is successful (in terms of achieving a relatively competitive validation error). We used the per-workload targets set by the ALGOPERF benchmark. These targets are useful because they provide a notion of a "competitive" value of the validation metrics for these specific workloads based on years of experience from the community. A definition of successful training based on such a well-founded notion of a "good" training result helps give us performance measures that are likely to spur useful algorithmic improvements on machine learning tasks.

**Performance Profile.** Aggregating the performance of a training algorithm over a heterogeneous set of workloads poses the challenge of appropriately balancing the different metrics and scales across different workloads. While the time to target metric alleviates this problem by making the metric of performance uniform, the problem of balanced aggregation remains. An alternative is to present an algorithms × workload sized table of performance metrics; however such tables are often hard to parse as the number of algorithms/workloads grows. The ALGOPERF benchmark adopted *performance profiles* (Dolan and Moré, 2002), which are a convenient measurement of the performance of a set of algorithms agregated across a set of workloads. For every algorithm, the performance profile plots the fraction of workloads where the given method's training time is within some ratio of the best per-workload training time. Specifically, given a set of training algorithms $S$ and workloads $W$, let $t_{s,w}$ for any $s \in S, w \in W$ be the time taken by the training algorithm $s$ on workload $w$. We can now define the fraction $r_{s,w}$ which is ratio of the given training algorithm $s$'s time on the workload and the training time of the best performing training algorithm on the same workload, i.e.

$$r_{s,w} = \frac{t_{s,w}}{\min_{s' \in S} t_{s',w}}.$$

We can now define the performance profile plot. For a given training algorithm $s$, the performance profile $p_s(\tau)$ for any $\tau \geqslant 1$ is defined as the fraction of workloads $w$, where the performance of $s$ is within a $\tau$ factor of the performance of the fastest algorithm on that workload. Mathematically,

$$p_s(\tau) = \frac{|w \in W : r_{s,w} \leqslant \tau|}{|W|}.$$

Thus for any training algorithm, a performance profile in a single plot provides us the information of the fraction of workloads where this training algorithm performs nearly as well as any other training algorithm tested on that workload, at various levels of tolerance. Note that the performance profile is by definition a function of the entire set of training algorithms scored.

**AlgoPerf Benchmark Score.** Having defined the notion of a performance profile $p_s(\tau)$, we can distill a single numerical *score* by considering the area under the performance curve (up to some maximum $\tau$). Intuitively, the more area covered by the performance profile of a training algorithm, the closer it performs to optimal (amongst the comparison set) on a larger number of workloads. The area covered by the performance profile curve is referred to as the ALGOPERF benchmark score, and is used to select final hyperparameter configurations (comparing different hyperparameters for a single algorithm) as well as to compare the best hyperparameter settings across algorithms. Since the benchmark score is based on the performance profiles it

is also dependent on the set of algorithms being compared. The benchmark score does not get credit for workloads where the target is never reached.

# 4 Naive learning-rate-free methods

We first considered the performance of the naive learning-rate-free methods—"out-of-the-box" without any additional warmup/annealing schedules. The reason for such an initial approach is that the goal of learning-rate-free methods is to leave any tuning of the learning rate (and therefore the schedule) to the algorithms. Namely, we took the default parameters of the algorithms as implemented by the authors in publicly available libraries, detailed in Table 1. For algorithms that employed additional momentum parameters (D-ADAPT, MECHANIC, PRODIGY, MoMo), we kept these momentum parameters as given in ADAM ($\beta_1 = 0.9, \beta_2 = 0.999$). Moreover, to fully test the ability of these algorithms to remove tuning, we considered no weight decay ($\lambda = 0$) on the parameters.

## 4.1 Naive learning-rate-free vs. Adam-derived baselines

All of the naive learning-rate-free algorithms performed poorly compared to our baselines across all workloads (Table 4). Additionally, only one algorithm reached any of the targets set by the ALGOPERF benchmark, which it did for just a single workload. Since so few workloads were trained successfully, analyzing the performance of these algorithms is not especially useful in the ALGOPERF framework of time-to-result scores.

We found no single, simple explanation for the poor performance of these naive learning-rate-free methods; detailed training curves can be found in Appendix B. Our results suggest that across a variety of dataset-model pairs, learning-rate-free methods have poor performance when using defaults from papers or open sourced codebases.

In contrast, evidence-based defaults allowed the simpler ADAMW and NADAMW baselines to perform well across many workloads. A natural question is whether or not a similar procedure to find better default hyperparameter settings for learning-rate-free algorithms—a question we explore in the next section.

Table 4: Best validation metrics over the course of training for baselines and naive learning-rate-free methods. Numbers colored in light gray indicate that the target was not achieved, while number colored in black indicate that the target was achieved. Validation metrics for naive learning-rate-free methods fail to beat NADAMW baselines on any workload (top row). Up arrow indicates metric is maximized, down arrow minimized.

| | Criteo 1TB | fastMRI | ImageNet | | LibriSpeech | | OGBG | WMT |
|---|---|---|---|---|---|---|---|---|
| | DLRMsmall | U-Net | ResNet-50 | ViT | Conformer | DeepSpeech | GNN | Transformer |
| Metric | CE ↓ | SSIM ↑ | Error Rate ↓ | Error Rate ↓ | WER ↓ | WER ↓ | mAP ↑ | BLEU ↑ |
| ADAMW | 0.1235 | 0.722 | 0.252 | 0.222 | 0.085 | 0.118 | **0.254** | 29.64 |
| NADAMW | **0.1235** | **0.723** | **0.226** | **0.214** | **0.079** | **0.109** | 0.253 | **30.07** |
| COCOB | 0.1238 | 0.719 | 0.305 | 0.274 | 0.172 | 0.149 | 0.234 | 26.67 |
| D-ADAPT (w. ADAM) | 0.1236 | 0.722 | 0.314 | 0.251 | 0.095 | 0.135 | 0.221 | 28.63 |
| DoG | 0.1334 | 0.714 | 0.317 | 0.275 | 0.467 | 0.226 | 0.231 | 26.34 |
| DoWG | 0.1259 | 0.715 | 0.389 | 0.998 | 0.844 | 0.839 | 0.205 | 0.00 |
| MECHANIC (w. ADAM) | 0.1240 | 0.722 | 0.321 | 0.249 | 0.098 | 0.124 | 0.235 | 1.82 |
| MECHANIC (w. NADAM) | 0.1241 | 0.721 | 0.318 | 0.248 | 0.099 | 0.123 | 0.239 | 4.50 |
| MECHANIC | 0.1361 | 0.714 | 0.301 | 0.263 | 0.183 | 0.209 | 0.045 | 11.94 |
| MoMo | 0.1254 | 0.716 | 0.313 | 0.268 | 0.116 | 0.154 | 0.236 | 25.78 |
| MoMo (w. ADAM) | 0.1240 | 0.723 | 0.313 | 0.999 | 0.368 | 0.454 | 0.221 | 0.99 |
| PRODIGY | 0.1241 | 0.723 | 0.317 | 0.251 | 0.098 | 0.149 | 0.212 | 0.57 |

# 5 AlgoPerf-calibrated learning-rate-free methods

The relative success of the evidence-based baselines over naive learning-rate-free algorithms suggest that tuning the additional optimization and regualarization hyperparameters is important for good cross-workload performance. We applied a similar search procedure to *calibrate* learning-rate-free methods to test their full potential as hyperparameter-free methods.

## 5.1 Calibrating learning-rate-free methods

We first conducted a quasi-random search for hyperparameters while using a training horizon of 50%, using 200 points (as described in Section 3.2). We chose to analyze MECHANIC, PRODIGY, DoG, MoMo and COCOB; for expediency, we omitted D-ADAPT and DoWG, given that PRODIGY is a successor to D-ADAPT, and DoWG is quite similar to DoG. For all algorithms with ADAM/NADAM variants, we used the ADAM variant to simplify our exploration. For algorithms which accepted relative schedules, we used a linear warmup + cosine decay schedule with a decay to the end of the 50% horizon.

In this initial search, only PRODIGY and MECHANIC (with ADAM) had any hyperparameter settings which were able to train more than 2 workloads successfully (Appendix D). Therefore, we focused the rest of our analysis on these two methods.

## 5.2 Analysis of AlgoPerf-calibrated methods

We calibrated PRODIGY and MECHANIC (with ADAM) using the full procedure described in Section 3.2—extending to additional training horizons and carrying out the final selection and benchmarking steps. In addition to the original 50% search, we added 33% and 66% horizons (200 points each). We took the putative top 3 hyperparameter settings (per algorithm per horizon) and subjected them to a more rigorous 5-seed evaluation to select the final, ALGOPERF-calibrated hyperparameters using the ALGOPERF benchmark score. This is exactly the same as the procedure for the baselines, except for the fact that the base learning rate was not tuned. If removing the learning rate is truly successful, by avoiding wasting trials that search bad learning rates, there should be effectively denser coverage of the other hyperparameters in the search as compared to the baselines.

Our calibration procedure greatly improved validation metrics. Table 5 shows the best validation metrics for the 3 hyperparameter points selected for each tuned algorithm at the 33%, 50% and 66% training horizons. AlgoPerf-calibrated versions of MECHANIC and PRODIGY now can train $4-5$ workloads successfully at multiple training horizons. The baselines no longer have the best scores on all workloads; in particular, the best validation metrics on IMAGENET and LIBRISPEECH DEEPSPEECH now come from learning-rate-free methods.

Since calibrated versions of PRODIGY and MECHANIC train successfully on more workloads, the ALGOPERF benchmark score for ranking the algorithms based on training speed becomes much more meaningful.

We used the ALGOPERF benchmark score (computed over all candidates collectively) to select a final representative for each algorithm-horizon pair (Table 6). On this leaderboard, PRODIGY and MECHANIC rank 3rd and 4th by AlgoPerf score. MECHANIC (with 33% horizon) reached fewer targets, but reached them faster to attain its high ranking, while PRODIGY (with 66% horizon) reached more targets somewhat more slowly to obtain its ranking. This result is reflected in the performance profiles (Figure 1). Note that MECHANIC with longer horizons did not train any additional workloads successfully, and PRODIGY with a smaller training horizon reached the target on fewer workloads. The much-improved cross-workload performance of MECHANIC and PRODIGY using evidence-based default hyperparameters suggests that it's possible for algorithm designers to provide more useful default settings for their algorithms—which may improve uptake within the broader research community.

Our results suggest that PRODIGY and MECHANIC, among all the learning-rate-free methods we tested, are the most promising as a step towards hyperparameter-free training algorithms. However, they are not demonstrably better than ADAMW and NADAMW at identical offline tuning budgets. In other words, once we selected a single hyperparameter configuration to use across all workloads, the fact that PRODIGY and

Mechanic did not need to include a default value for the base learning rate in the configuration did not provide a detectable advantage over AdamW or NadamW. Although they are designed to adapt the overall learning rate per-workload, in our results, removing the overall base learning rate from the tuning problem did not actually seem to make the search for a configuration of the other non-learning rate hyperparameters that worked well across multiple workloads appreciably easier. Any benefits from removing the learning rate were not strong enough to take Prodigy and Mechanic to the top of the leaderboard, let alone get the top spot in a convincing manner.

Table 5: Best validation metric values for AlgoPerf-calibrated methods achieved over the course of training. Numbers colored in light gray indicate that the target was not achieved, while number colored in black indicate that the target was achieved. Validation metrics for learning-rate-free methods with calibrated regularization and "decay schedule" are competitive with AdamW and NadamW baselines. Validation scores are much closer to baseline for all algorithms, and even beat the baseline on certain workloads.

| | Criteo 1TB | fastMRI | ImageNet | | LibriSpeech | | OGBG | WMT |
|---|---|---|---|---|---|---|---|---|
| | DLRMsmall | U-Net | ResNet-50 | ViT | Conformer | DeepSpeech | GNN | Transformer |
| Metric | CE ↓ | SSIM ↑ | Error Rate ↓ | Error Rate ↓ | WER ↓ | WER ↓ | mAP ↑ | BLEU ↑ |
| AdamW 33% | 0.1235 | 0.722 | 0.252 | 0.222 | 0.085 | 0.118 | 0.254 | 29.64 |
| AdamW 50% | 0.1235 | 0.724 | 0.277 | 0.531 | 0.100 | 0.118 | 0.187 | 24.01 |
| AdamW 66% | 0.1234 | 0.722 | 0.234 | 0.236 | 0.075 | 0.108 | 0.257 | 29.60 |
| NadamW 33% | 0.1236 | 0.723 | 0.252 | 0.217 | 0.106 | 0.116 | 0.250 | 29.64 |
| NadamW 50% | 0.1234 | 0.722 | 0.251 | 0.232 | 0.085 | 0.115 | 0.276 | 30.85 |
| NadamW 66% | 0.1235 | 0.723 | 0.226 | 0.214 | 0.079 | 0.109 | 0.253 | 30.07 |
| Mechanic (w. Adam) 33% | 0.1240 | 0.724 | 0.238 | 0.225 | 0.084 | 0.113 | 0.245 | 29.91 |
| Mechanic (w. Adam) 50% | 0.1234 | 0.723 | 0.245 | 0.213 | 0.085 | 0.112 | 0.239 | 2.59 |
| Mechanic (w. Adam) 66% | 0.1235 | 0.723 | 0.275 | 0.222 | 0.082 | 0.111 | 0.245 | 25.53 |
| Prodigy (w. Adam) 33% | 0.1239 | 0.723 | 0.239 | 0.336 | 0.082 | 0.117 | 0.198 | 30.11 |
| Prodigy (w. Adam) 50% | 0.1234 | 0.722 | 0.223 | 0.213 | 0.087 | 0.107 | 0.244 | 29.94 |
| Prodigy (w. Adam) 66% | 0.1236 | 0.721 | 0.223 | 0.208 | 0.085 | 0.103 | 0.220 | 28.81 |

Table 6: AlgoPerf-calibrated methods show mixed promise in terms of time-to-target metrics. Biggest improvements are on ImageNet ViT and LibriSpeech Conformer workloads. It is important to reemphasize that the score of these representative examples is a function of the entire algorithm pool, including the points that were not selected and suppressed from Table 6. The scores and time-to-target ratios of the complete set of algorithms that was used to compute the scores can be found in Appendix E.

| | AlgoPerf | Criteo 1TB | fastMRI | ImageNet | | LibriSpeech | | OGBG | WMT |
|---|---|---|---|---|---|---|---|---|---|
| | Score | DLRMsmall | U-Net | ResNet-50 | ViT | Conformer | DeepSpeech | GNN | Transformer |
| AdamW 33% | 0.498 | **0.284** | > 1 | > 1 | 0.303 | 0.245 | 0.291 | > 1 | > 1 |
| NadamW 66% | 0.480 | 0.429 | > 1 | 0.782 | 0.609 | 0.435 | 0.501 | > 1 | > 1 |
| Prodigy (w. Adam) 66% | 0.477 | 0.470 | > 1 | 0.773 | 0.521 | 0.491 | 0.500 | > 1 | > 1 |
| Mechanic (w. Adam) 33% | 0.469 | > 1 | **0.130** | > 1 | 0.335 | 0.248 | 0.303 | > 1 | > 1 |
| Prodigy (w. Adam) 50% | 0.455 | 0.388 | > 1 | **0.577** | 0.404 | > 1 | 0.381 | > 1 | > 1 |
| NadamW 50% | 0.452 | 0.335 | > 1 | > 1 | > 1 | 0.369 | 0.412 | > 1 | **0.457** |
| Mechanic (w. Adam) 50% | 0.434 | 0.382 | > 1 | > 1 | 0.412 | 0.375 | 0.374 | > 1 | > 1 |
| Mechanic (w. Adam) 66% | 0.375 | 0.427 | > 1 | > 1 | 0.608 | 0.436 | 0.478 | > 1 | > 1 |
| NadamW 33% | 0.369 | 0.320 | > 1 | > 1 | **0.303** | > 1 | **0.286** | > 1 | > 1 |
| AdamW 66% | 0.276 | 0.511 | > 1 | > 1 | > 1 | 0.419 | 0.518 | > 1 | > 1 |
| Prodigy (w. Adam) 33% | 0.245 | > 1 | > 1 | > 1 | > 1 | **0.242** | 0.317 | > 1 | > 1 |
| AdamW 50% | 0.242 | 0.489 | 0.229 | > 1 | > 1 | > 1 | 0.492 | > 1 | > 1 |

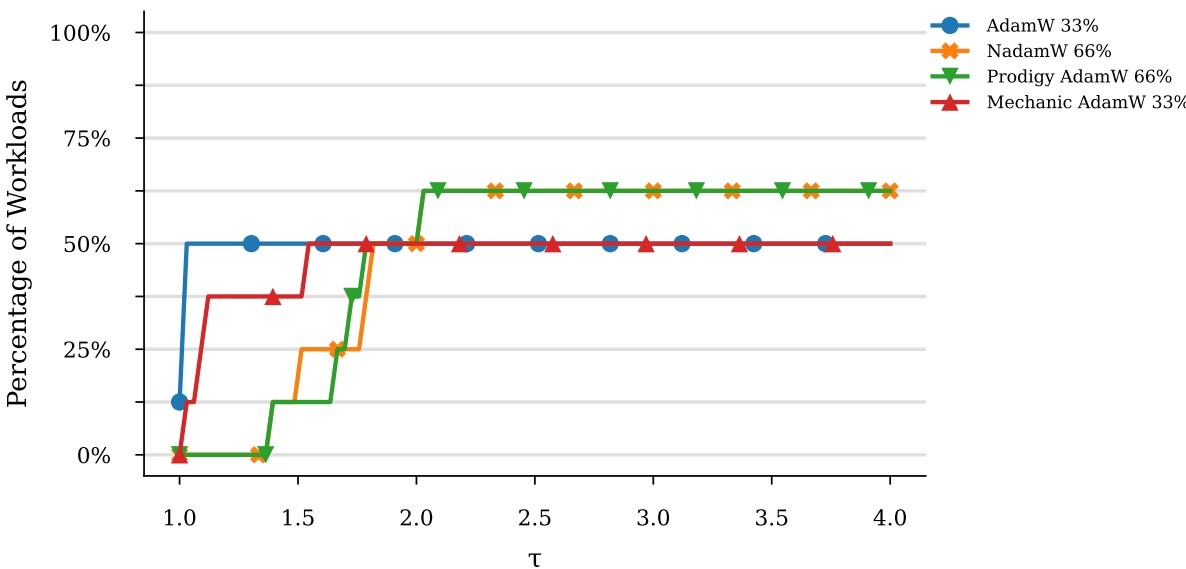

Figure 1: Performance profiles of top 4 algorithms from Table 6. These curves represent the best performances, by AlgoPerf benchmark score, for AdamW, NadamW, Prodigy, and Mechanic over 3 different horizons and 3 hyperparameter settings per algorithm-horizon pair from the broad search. AdamW and Prodigy obtain their high scores by hitting targets for 4 workloads relatively quickly. NadamW and Prodigy hit 5 targets, but more slowly to obtain their scores.

## 6 Discussion

Our experiments represent the first comprehensive study of how well learning-rate-free methods generalize across workloads, and whether they actually provide a practical benefit in terms of simplifying the overall hyperparameter tuning problem. Our experiments address two important measurement pitfalls: comparing a learning-rate-free method to (say) Adam using a carefully tuned learning rate *that is tuned per-workload* provides an overly pessimistic assessment, but comparing to a standard optimizer using library defaults is overly optimistic. Instead, we only compared to baselines that also did no workload-specific tuning, which also lets us assess the potential role of learning-rate-free training algorithms as a component of future hyperparameter-free training algorithms. By using the AlgoPerf benchmark to ground our calibration procedure, our characterization of the performance of learning-rate-free methods reflects a diverse set of realistic deep learning workloads and a meaningful training success criterion (based on reaching a good validation performance). As a result, our calibration procedure found evidence-based default hyperparameter settings that dramatically improved upon the library and paper defaults we started with.

We focused on the "hyperparameter-free" comparison setting because learning-rate-free methods are the single most studied hyperparameter reduction technique in deep learning. The primary motivation of learning-rate-free methods is that they can select good learning rates per workload, and even per individual step of training; a natural followup question, then, is whether or not this capability actually leads to better cross-workload performance without re-tuning. We found that the "default" values reported in the literature, or found in standard implementations, did not lead to good performance on most workloads.

Indeed, we had to tune over $\beta_1$ and $\beta_2$ for Adam derived methods, as well as the regularization parameters to obtain AlgoPerf-calibrated methods that hit a majority of the validation targets in the allotted training budget. This result suggests that developers of new algorithms would be well-served by finding calibrated, evidence-based defaults and publishing them in papers and open source code. Such a process should include tuning of regularization parameters, which can have optimizer-dependent interactions. Better justified default values could improve adoption of these methods, and could speed up overall research in the field.

The learning-rate-free methods studied here avoided the need to tune the *base* learning rate only. Another hidden tuning parameter in learning-rate-free methods is the training horizon, or more generally, the learning rate schedule. All the best performing methods we examined in this work still needed some kind of relative learning rate schedule with warmup and decay. In this sense, they are not truly learning-rate-free. A recent algorithm proposed in Defazio et al. (2024) attempts to remove the learning rate schedule; the resulting SCHEDULE-FREE ADAM convincingly won the self-tuning track in the recent ALGOPERF self-tuning public competition (Kasimbeg et al., 2025). In Appendix F we make a best-effort comparison between the competition version of SCHEDULE-FREE ADAM and find that its estimated benchmark score is better than any method obtained by our evidence based search, in large part because of its ability to hit 7 out of the 8 validation targets (compared to the maximum of 5 from algorithms in our search). This suggests that successfully removing the learning rate schedule may be just as important as removing the need for a base learning rate.

In the end, even the best performing ALGOPERF-calibrated learning-rate-free algorithms performed slightly worse than the best ADAMW/NADAMW baselines—even though the same number of potential configurations were searched to produce them. This result suggests that the learning-rate-free methods we tested don't yet save time on hyperparameter tuning, and don't provide significant cross-workload generalization over more conventional methods.

One limitation of our study is the selection of workloads. While we believe that ALGOPERF covers a broad, well-motivated set of problems, some effects can't be tested using the setup. In particular, many practical settings now involve training models at a variety of scales, to extrapolate good hyperparameters for training a final, massive model (Hoffmann et al., 2022). We know theoretically and empirically that most simple training algorithms require scale-specific values for hyperparameters (Lee et al., 2019). In these settings, calibrated hyperparameters as we've defined them do not exist without techniques like dimensionally-aware parameterizations or scaling heuristics for hyperparameters (Everett et al., 2024). Incorporating these additional considerations could shed more light on the promise of learning-rate-free methods as the basis for hyperparameter-free training algorithms.

Our analysis identified PRODIGY and MECHANIC as the two most promising methods among those we tested. We believe that our experimental approach should be emulated in future work proposing new methods and provides an example of how to more rigorously test the efficacy of learning-rate-free methods. Though none of the learning-rate-free methods we tried are yet ready to supplant standard optimizers, we believe the future is bright for the development of such adaptive optimizers, now that our benchmarking and evaluation capabilities have improved.

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

## A How to tune Parameter-Free algorithms?

Unfortunately, the recent proliferation of Learning-rate free methods Mishchenko and Defazio (2023); Defazio and Mishchenko (2023); Defazio et al. (2023); Cutkosky et al. (2023); Ivgi et al. (2023); Orabona and Tommasi (2017) are still far from being parameter-free methods. All of these papers tackle the learning-rate hyperparameter in neural network training, leaving no or *default* prescriptions to be used for other hyperparameters for eg. first and second moment accumulation hyperparameters ($\beta_1, \beta_2$ in Adam-like methods), weight decay and other regularization hyperparameters such as label smoothing and dropout. While for well-established and repeatedly trained on workloads in deep learning a usable value for these parameters can be found in literature however those values are unlikely to be usable in practice more generally. Further while these algorithms tackle the learning rate scalar, they often employ a learning rate schedule to reach good performance in deep learning tasks, thereby not making the learning rate fully parameter-free as well. Learning rate schedules need to be treated as a multi-dimensional hyperparameter in itself. Even when you fix the rough 'shape' of the learning rate schedule, for example for the commonly used *warmup+cosine decay* schedule, parameters like how many steps (or what percentage of training) of *warmup* is performed or when does the decay end still needs to be determined.

We consider the following set of recently proposed *parameter-free* algorithms in this context

- D-ADAPT Defazio and Mishchenko (2023)

- MECHANIC Cutkosky et al. (2023)

- PRODIGY Mishchenko and Defazio (2023)

- DOG Ivgi et al. (2023)

We perform the following experiments on these algorithms

1. **No tuning:** We run the presented algorithms without any modification from their default values as found in the Optax implementations. For the regularization hyperparameters, since there are no well-defined values, we choose the default values supplied, which effectively, turns these parameters off (i.e. dropout rates, label smoothing strength, and weight decay of 0). Note that this also means we do not provide an external learning rate schedule and the only schedule applied if any is maintained by algorithm. To provide a point of comparison to these algorithms, we ran experiments with ADAM and NADAM with default hyperparameters (and the same default regularization hyperparameters) with an additional sweep over the (constant) learning rate.

2. **Tuning other hyperparameters** In this section we use the above algorithms with their default optimization hyperparameters and tune the following regularization and LR schedule parameters on top of it

   - Weight Decay - [1e-1, 1e-2, 1e-3, 1e-4, 1e-5, 1e-6, 1e-7]
   - Dropout (tied) - [0.0, 0.1]
   - Label Smoothing - [0.0, 0.1, 0.2]
   - Training Horizon - [0.33, 0.49, 0.64, 0.79, 1.0]
   - Warmup Fraction - [0.02, 0.05, 0.10]

## B    Naive learning-rate-free methods training curves

### Criteo1TB

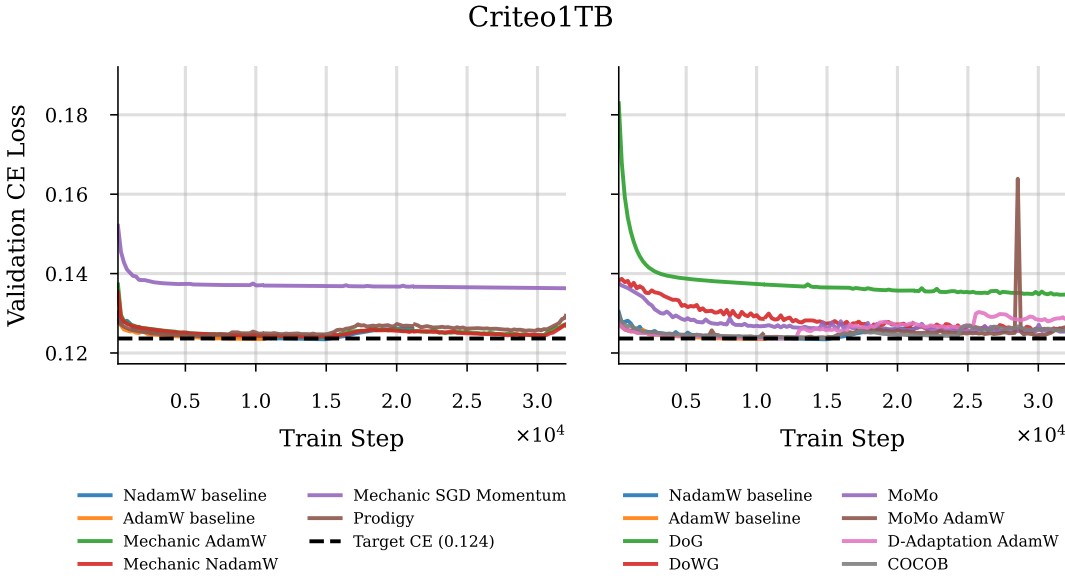

Figure 2: Validation CE loss vs training step for 'out-of-the-box' learning-rate-free methods and baselines on CRITEO 1TB DLRM SMALL.

### FastMRI

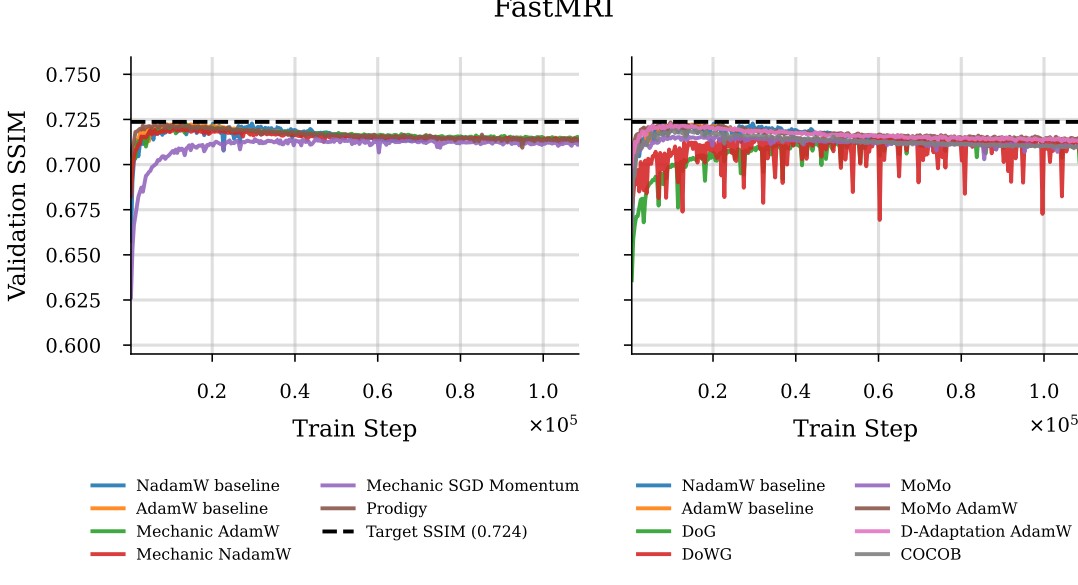

Figure 3: Validation SSIM vs training step for 'out-of-the-box' learning-rate-free-methods and baselines on FASTMRI.

Imagenet ResNet50

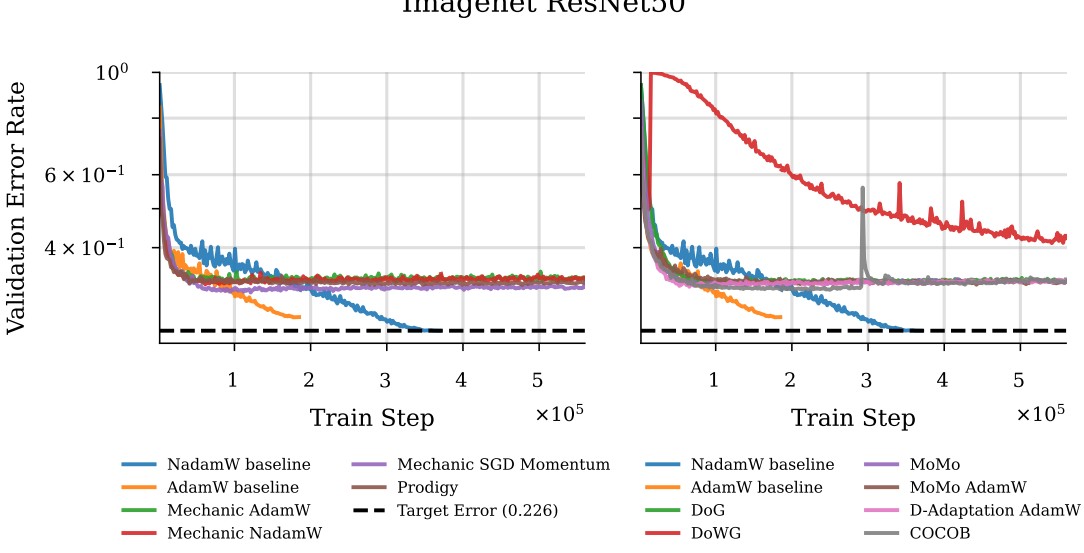

Figure 4: Validation Error Rate vs train step for 'out-of-the-box' learning-rate-free methods and baselines on IMAGENET RESNET-50.

Imagenet ViT

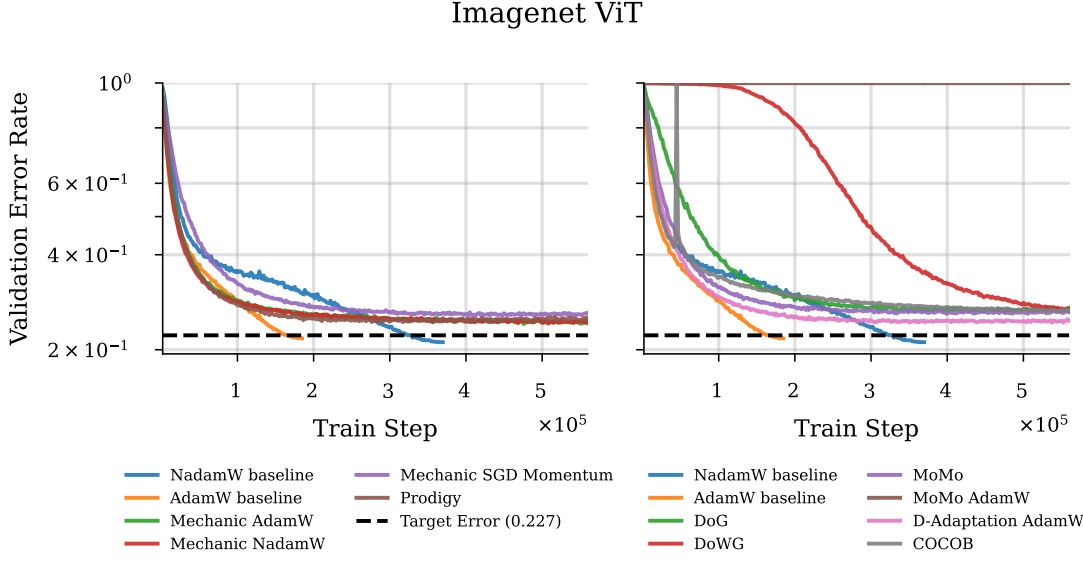

Figure 5: Validation Error Rate vs train step for 'out-of-the-box' learning-rate-free methods and baselines on IMAGENET VIT.

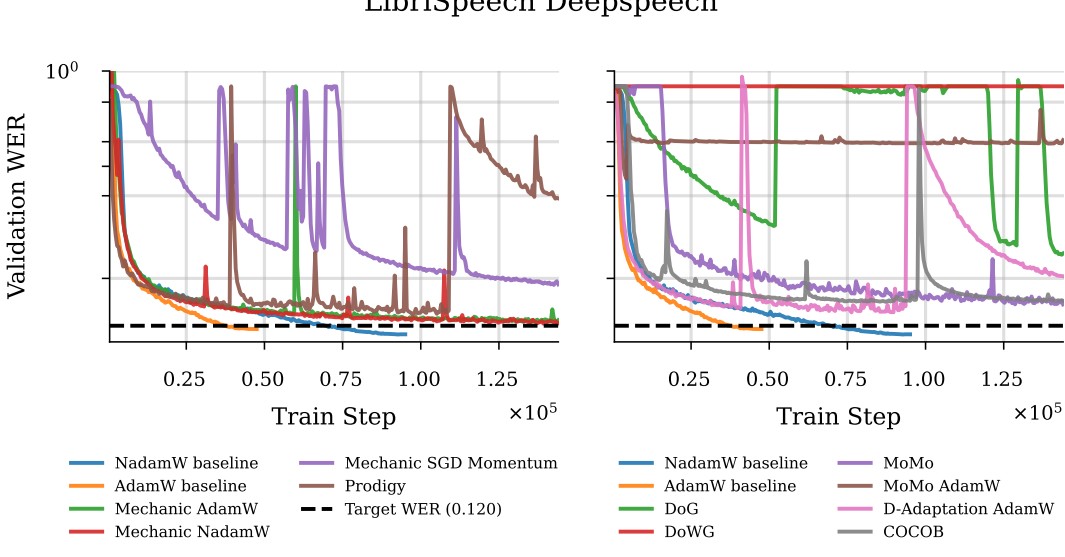

Figure 6: Validation WER vs train step for 'out-of-the-box' learning-rate-free methods and baselines on LIBRISPEECH CONFORMER.

Figure 7: Validation WER vs train step for 'out-of-the-box' learning-rate-free methods and baselines on LIBRISPEECH DEEPSPEECH.

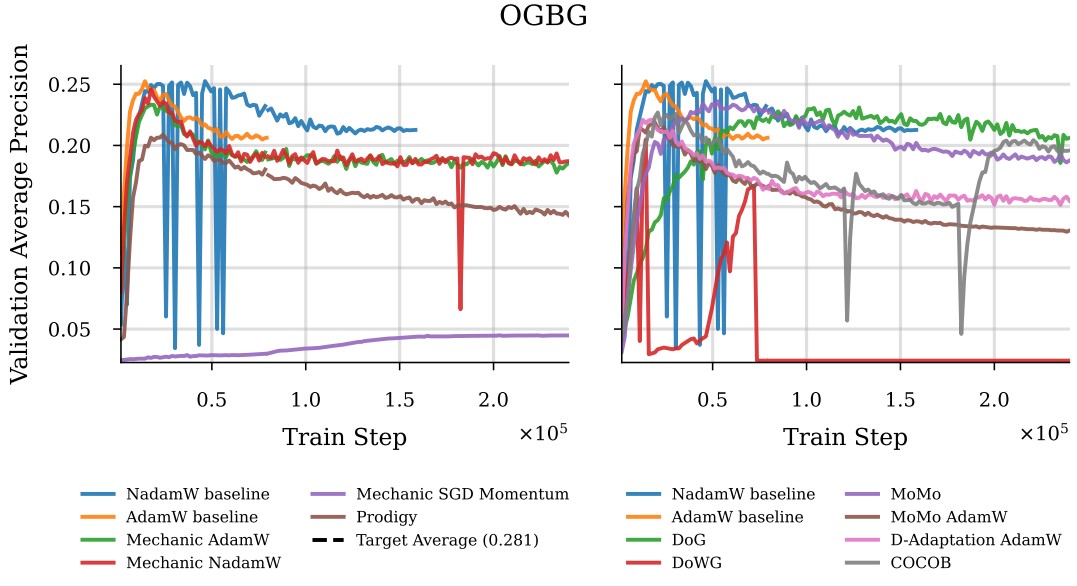

Figure 8: Validation Average Precision vs train step for 'out-of-the-box' learning-rate-free methods and baselines on OGBG GNN.

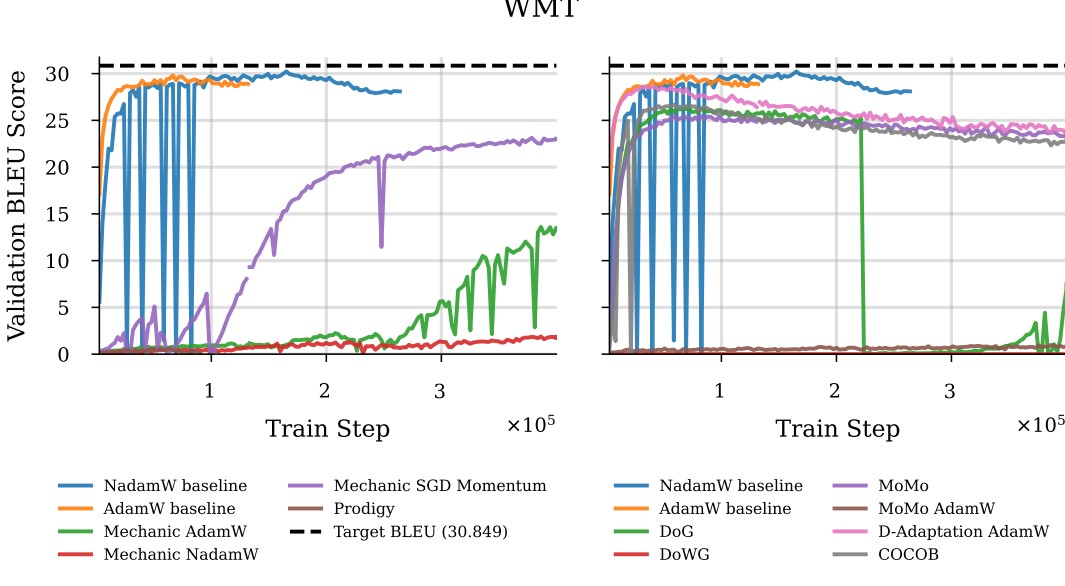

Figure 9: Validation BLEU score vs train step for 'out-of-the-box' learning-rate-free methods and baselines on WMT Transformer.

## C  Baseline selection

Table 7: ALGOPERF scores and runtime fractions of the baseline candidate methods. We chose the highest scoring NADAMW and ADAMW methods as the baselines. These baseline rows are colored in gray. Note that these methods were scored in a wider population of methods. The scores and runtime fractions of the full population are given in Table 9.

| | AlgoPerf Score | Criteo 1TB DLRMsmall | fastMRI U-Net | ImageNet ResNet-50 | ViT | LibriSpeech Conformer | DeepSpeech | OGBG GNN | WMT Transformer |
|---|---|---|---|---|---|---|---|---|---|
| NADAMW 33% | 0.248 | > 1 | > 1 | > 1 | 0.317 | > 1 | 0.282 | > 1 | > 1 |
| NADAMW 33% | 0.369 | 0.320 | > 1 | > 1 | 0.303 | > 1 | 0.286 | > 1 | > 1 |
| NADAMW 33% | 0.248 | > 1 | > 1 | > 1 | 0.306 | > 1 | 0.296 | > 1 | > 1 |
| NADAMW 50% | 0.452 | 0.335 | > 1 | > 1 | > 1 | 0.369 | 0.412 | > 1 | 0.457 |
| NADAMW 50% | 0.356 | > 1 | > 1 | 0.566 | > 1 | > 1 | 0.414 | 0.329 | > 1 |
| NADAMW 50% | 0.101 | > 1 | > 1 | > 1 | > 1 | > 1 | 0.442 | > 1 | > 1 |
| NADAMW 66% | 0.125 | > 1 | 0.084 | > 1 | > 1 | > 1 | > 1 | > 1 | > 1 |
| **NadamW 66%** | **0.480** | 0.429 | > 1 | 0.782 | 0.609 | 0.435 | 0.501 | > 1 | > 1 |
| NADAMW 66% | 0.000 | > 1 | > 1 | > 1 | > 1 | > 1 | > 1 | > 1 | > 1 |
| ADAMW 33% | 0.248 | 0.298 | > 1 | > 1 | > 1 | 0.244 | > 1 | > 1 | > 1 |
| **AdamW 33%** | **0.498** | 0.284 | > 1 | > 1 | 0.303 | 0.245 | 0.291 | > 1 | > 1 |
| ADAMW 33% | 0.125 | 0.284 | > 1 | > 1 | > 1 | > 1 | > 1 | > 1 | > 1 |
| ADAMW 50% | 0.120 | > 1 | 0.095 | > 1 | > 1 | > 1 | > 1 | > 1 | > 1 |
| ADAMW 50% | 0.000 | > 1 | > 1 | > 1 | > 1 | > 1 | > 1 | > 1 | > 1 |
| ADAMW 50% | 0.242 | 0.489 | 0.229 | > 1 | > 1 | > 1 | 0.492 | > 1 | > 1 |
| ADAMW 66% | 0.272 | > 1 | > 1 | > 1 | 0.577 | 0.453 | 0.478 | > 1 | > 1 |
| ADAMW 66% | 0.276 | 0.511 | > 1 | > 1 | > 1 | 0.419 | 0.518 | > 1 | > 1 |
| ADAMW 66% | 0.000 | > 1 | > 1 | > 1 | > 1 | > 1 | > 1 | > 1 | > 1 |

## D  Max Targets Hit in Initial Search on 50% Training Horizon

Table 8: Targets hit per algorithm in initial search. For each class of algorithms we chose a representative algorithm and train with 100 hyperparameter points sampled from the initial search space on a 50% training horizon. Only PRODIGY and MECHANIC (with ADAM) had any hyperparameter settings which were able to train more than 2 workloads successfully.

| Algorithm Class Members | Algorithm | Max Workload Targets Hit |
|---|---|---:|
| MECHANIC | MECHANIC | 4 |
| PRODIGY, D-ADAPT | PRODIGY | 4 |
| DoG, DoWG | DoG | 0 |
| MoMo | MoMo | 2 |
| COCOB | COCOB | 0 |

# E  AlgoPerf metrics for entire set of methods scored

Schedule-Free Adam was intr

Table 9: AlgoPerf scores and runtime fractions of entire set of methods that account for the performance profiles and AlgoPerf scores. Methods that were suppressed from table 6 are indicated with gray rows. These methods were suppressed if there exist a similar algorithm in the same class of algorithm and training horizon that outperformed the method in terms of AlgoPerf score. Bold numbers indicate the smallest runtime fraction over all submission on a given workload.

| | AlgoPerf | Criteo 1TB | fastMRI | ImageNet | | LibriSpeech | | OGBG | WMT |
|---|---|---|---|---|---|---|---|---|---|
| | Score | DLRMsmall | U-Net | ResNet-50 | ViT | Conformer | DeepSpeech | GNN | Transformer |
| AdamW 33% [1] | 0.498 | **0.284** | > 1 | > 1 | 0.303 | 0.245 | 0.291 | > 1 | > 1 |
| NadamW 66% [1] | 0.480 | 0.429 | > 1 | 0.782 | 0.609 | 0.435 | 0.501 | > 1 | > 1 |
| Prodigy (w. Adam) 66% [2] | 0.477 | 0.470 | > 1 | 0.773 | 0.521 | 0.491 | 0.500 | > 1 | > 1 |
| Mechanic (w. Adam) 33% [0] | 0.469 | > 1 | 0.130 | > 1 | 0.335 | 0.248 | 0.303 | > 1 | > 1 |
| Prodigy (w. Adam) 50% [2] | 0.455 | 0.388 | > 1 | 0.577 | 0.404 | > 1 | 0.381 | > 1 | > 1 |
| NadamW 50% [0] | 0.452 | 0.335 | > 1 | > 1 | > 1 | 0.369 | 0.412 | > 1 | **0.457** |
| Mechanic (w. Adam) 50% [2] | 0.434 | 0.382 | > 1 | > 1 | 0.412 | 0.375 | 0.374 | > 1 | > 1 |
| Mechanic (w. Adam) 50% [1] | 0.399 | 0.513 | > 1 | > 1 | 0.486 | 0.346 | 0.448 | > 1 | > 1 |
| Mechanic (w. Adam) 66% [0] | 0.375 | 0.427 | > 1 | > 1 | 0.608 | 0.436 | 0.478 | > 1 | > 1 |
| NadamW 33% [1] | 0.369 | 0.320 | > 1 | > 1 | **0.303** | > 1 | 0.286 | > 1 | > 1 |
| NadamW 50% [1] | 0.356 | > 1 | > 1 | **0.566** | > 1 | > 1 | 0.414 | **0.329** | > 1 |
| Mechanic (w. Adam) 50% [0] | 0.308 | 0.450 | > 1 | > 1 | 0.479 | > 1 | 0.405 | > 1 | > 1 |
| Mechanic (w. Adam) 66% [1] | 0.286 | > 1 | > 1 | > 1 | 0.533 | 0.433 | 0.446 | > 1 | > 1 |
| AdamW 66% [1] | 0.276 | 0.511 | > 1 | > 1 | > 1 | 0.419 | 0.518 | > 1 | > 1 |
| AdamW 66% [0] | 0.272 | > 1 | > 1 | > 1 | 0.577 | 0.453 | 0.478 | > 1 | > 1 |
| NadamW 33% [0] | 0.248 | > 1 | > 1 | > 1 | 0.317 | > 1 | **0.282** | > 1 | > 1 |
| NadamW 33% [2] | 0.248 | > 1 | > 1 | > 1 | 0.306 | > 1 | 0.296 | > 1 | > 1 |
| AdamW 33% [0] | 0.248 | 0.298 | > 1 | > 1 | > 1 | 0.244 | > 1 | > 1 | > 1 |
| Prodigy (w. Adam) 33% [1] | 0.245 | > 1 | > 1 | > 1 | > 1 | **0.242** | 0.317 | > 1 | > 1 |
| AdamW 50% [2] | 0.242 | 0.489 | 0.229 | > 1 | > 1 | > 1 | 0.492 | > 1 | > 1 |
| Mechanic (w. Adam) 33% [1] | 0.242 | > 1 | > 1 | > 1 | 0.348 | > 1 | 0.292 | > 1 | > 1 |
| Mechanic (w. Adam) 66% [2] | 0.232 | > 1 | > 1 | > 1 | 0.655 | 0.494 | 0.630 | > 1 | > 1 |
| Prodigy (w. Adam) 50% [1] | 0.214 | > 1 | > 1 | > 1 | > 1 | 0.323 | 0.429 | > 1 | > 1 |
| Prodigy (w. Adam) 50% [0] | 0.204 | > 1 | > 1 | > 1 | 0.457 | > 1 | 0.450 | > 1 | > 1 |
| Prodigy (w. Adam) 33% [2] | 0.181 | > 1 | 0.218 | > 1 | > 1 | > 1 | 0.303 | > 1 | > 1 |
| NadamW 66% [0] | 0.125 | > 1 | **0.084** | > 1 | > 1 | > 1 | > 1 | > 1 | > 1 |
| AdamW 33% [2] | 0.125 | **0.284** | > 1 | > 1 | > 1 | > 1 | > 1 | > 1 | > 1 |
| Mechanic (w. Adam) 33% [2] | 0.125 | > 1 | > 1 | > 1 | 0.303 | > 1 | > 1 | > 1 | > 1 |
| AdamW 50% [0] | 0.120 | > 1 | 0.095 | > 1 | > 1 | > 1 | > 1 | > 1 | > 1 |
| NadamW 50% [2] | 0.101 | > 1 | > 1 | > 1 | > 1 | > 1 | 0.442 | > 1 | > 1 |
| Prodigy (w. Adam) 66% [1] | 0.098 | > 1 | > 1 | > 1 | 0.502 | > 1 | > 1 | > 1 | > 1 |
| Prodigy (w. Adam) 66% [0] | 0.091 | > 1 | > 1 | > 1 | 0.548 | > 1 | > 1 | > 1 | > 1 |
| NadamW 66% [2] | 0.000 | > 1 | > 1 | > 1 | > 1 | > 1 | > 1 | > 1 | > 1 |
| AdamW 66% [2] | 0.000 | > 1 | > 1 | > 1 | > 1 | > 1 | > 1 | > 1 | > 1 |
| Prodigy (w. Adam) 33% [0] | 0.000 | > 1 | > 1 | > 1 | > 1 | > 1 | > 1 | > 1 | > 1 |
| AdamW 50% [1] | 0.000 | > 1 | > 1 | > 1 | > 1 | > 1 | > 1 | > 1 | > 1 |

## F    Adding Schedule-free Adam

SCHEDULE-FREE ADAM introduced by Defazio et al. (2024) achieved first place on the self-tuning track of the inaugural ALGOPERF competition Kasimbeg et al. (2025). At the time of writing, a JAX implementation of SCHEDULE-FREE ADAM capable of reproducing the runtime performance observed in the ALGOPERF competition was unavailable. Consequently, we were unable produce a ALGOPERF-calibrated SCHEDULE-FREE ADAM algorithm using the evidence-based search procedure described in 3.2.

To provide a good-faith comparison, we evaluated the SCHEDULE-FREE ADAM submission against our ALGOPERF-calibrated algorithms by extracting the runtime fractions of workload budgets from the the results of the ALGOPERF competition. Using the runtime fractions, we recomputed the ALGOPERF scores with SCHEDULE-FREE ADAM in the pool of submissions. Since the runtime budgets in our experimental setup are calibrated for the ALGOPERF setup we believe the fractions are good enough approximations for cross-platform comparisons. Note that this remains an approximation and that in an ideal setup the SCHEDULE-FREE ADAM algorithm would have been evaluated on the same hardware as the rest of the algorithms in the scoring pool.

The results show that SCHEDULE-FREE ADAM beats MECHANIC, PRODIGY and our baselines by a large margin in ALGOPERF score. The SCHEDULE-FREE ADAM reaches the target on 7/8 workloads, whereas the other algorithms reached the targets on a maximum of 5 workloads. The SCHEDULE-FREE ADAM algorithm also achieved the fastest runtime on 5 of the workloads.

| | AlgoPerf | Criteo 1TB | fastMRI | ImageNet | | LibriSpeech | | OGBG | WMT |
|---|---|---|---|---|---|---|---|---|---|
| | Score | DLRMsmall | U-Net | ResNet-50 | ViT | Conformer | DeepSpeech | GNN | Transformer |
| SCHEDULE-FREE ADAM | 0.860 | **0.249** | **0.050** | $> 1$ | **0.225** | 0.322 | 0.294 | **0.108** | **0.314** |
| ADAMW 33% | 0.478 | 0.284 | $> 1$ | $> 1$ | 0.303 | **0.245** | **0.291** | $> 1$ | $> 1$ |
| PRODIGY (W. ADAM) 66% | 0.443 | 0.470 | $> 1$ | 0.773 | 0.521 | 0.491 | 0.500 | $> 1$ | $> 1$ |
| NADAMW 66% | 0.442 | 0.429 | $> 1$ | 0.782 | 0.609 | 0.435 | 0.501 | $> 1$ | $> 1$ |
| PRODIGY (W. ADAM) 50% | 0.428 | 0.388 | $> 1$ | 0.577 | 0.404 | $> 1$ | 0.381 | $> 1$ | $> 1$ |
| NADAMW 50% | 0.426 | 0.335 | $> 1$ | $> 1$ | $> 1$ | 0.369 | 0.412 | $> 1$ | 0.457 |
| MECHANIC (W. ADAM) 33% | 0.409 | $> 1$ | 0.130 | $> 1$ | 0.335 | 0.248 | 0.303 | $> 1$ | $> 1$ |
| MECHANIC (W. ADAM) 50% | 0.407 | 0.382 | $> 1$ | $> 1$ | 0.412 | 0.375 | 0.374 | $> 1$ | $> 1$ |
| NADAMW 33% | 0.348 | 0.320 | $> 1$ | $> 1$ | 0.303 | $> 1$ | 0.286 | $> 1$ | $> 1$ |
| MECHANIC (W. ADAM) 66% | 0.337 | 0.427 | $> 1$ | $> 1$ | 0.608 | 0.436 | 0.478 | $> 1$ | $> 1$ |
| ADAMW 66% | 0.266 | 0.511 | $> 1$ | $> 1$ | $> 1$ | 0.419 | 0.518 | $> 1$ | $> 1$ |
| PRODIGY (W. ADAM) 33% | 0.245 | $> 1$ | $> 1$ | $> 1$ | $> 1$ | 0.242 | 0.317 | $> 1$ | $> 1$ |
| ADAMW 50% | 0.179 | 0.489 | 0.229 | $> 1$ | $> 1$ | $> 1$ | 0.492 | $> 1$ | $> 1$ |

Table 10: SCHEDULE-FREE ADAM outperforms the ALGOPERF-calibrated learning-rate-free algorithms and baselines.

## G    A brief history of learning-rate-free methods

### G.1    Linesearches in deterministic setting

The design of hyperparameter-free methods naturally depends on the algorithm itself and the objective at hand. For simple objectives such as convex quadratics, there exist optimal algorithms that are hyperparameter free such as the conjugate gradient method (Hestenes et al., 1952). However, even for simple convex objectives, optimization algorithms have generally been designed given known properties of the objective such as the smoothness or the strong convexity of the objective, which translates in the selection of some hyperparameters (Nesterov, 2018).

To illustrate the development of hyperparameter-free methods, we focus on a simple gradient descent. We consider the non-stochastic case to start with. A gradient descent requires the selection of a learning rate, often called stepsize in the optimization literature. To select the stepsize, several criterions have been proposed.

The first and most common criterion has been to enforce a *sufficient decrease* of the objective (Armijo, 1966). Namely, the stepsize $\eta$ is selected such that $f(w_k + \eta u_k) \leqslant f(w_k) + c_1 \eta \nabla f(w_k)^\top u_k$ for $u_k = -\nabla f(w_k)$, $c_1$ some small constant (of the order of $10^{-4}$ for example (Nocedal and Wright, 1999)), and $w_k$ the current parameters. The stepsize $\eta$ is then found by means of a while loop that decreases the stepsize by some constant factor until the aforementioned criterion is satisfied. Equipped with such a criterion a simple gradient descent has, up to logarithmic factors, the same convergence guarantees as with known smooth and strong convexity constants (Nesterov, 2018).

Ideally, we may rather want to select the stepsize to minimize the function along the update direction. Such an endeavor is generally too computationally expansive to be undertaken. To circumvent this issue, Wolfe (1969) proposed an additional *curvature condition* of the form $-u_k^\top \nabla f(w_k + \eta u_k) \leqslant -c_2 u_k^\top \nabla f(w_k)$. Such a curvature condition ensures that the selected stepsize is close to a local minimum of $\eta \to f(w_k + \eta u_k)$. Blending both sufficient and curvature conditions in an algorithm can be done by some form of dichotomy that carefully accounts for the geometry of the search (Moré and Thuente, 1994; Nocedal and Wright, 1999).

Additional linesearches have been proposed using for example an approximate sufficient decrease condition (Hager and Zhang, 2005) or alternative measurements of the curvature like Goldstein conditions (Nocedal and Wright, 1999). All aforementioned linesearches can be implemented as long as the update direction $u_k$ is a descent direction (that is $u_k^\top \nabla f(w_k) < 0$). In particular, these linesearch methods are an important ingredient for the implementation of quasi-newton methods like LBFGS that precondition the gradient with an estimate of the inverse Hessian (Nocedal and Wright, 1999).

## H  Detailed algorithms

We detail below in pseudocode the algorithms based on an estimation of the distance to the solution that we consider in our benchmark. D-ADAPT tailored for the ADAM optimizer is presented in Algorithm 1. PRODIGY is presented in Algorithm 2.

---

**Algorithm 1:** ADAM with D-ADAPT

---

1: **Inputs:**
- Initial parameters $w_0$,
- Initial estimate of the distance $d_0$ (default $10^{-6}$),
- Optional sequence of learning rate (to include e.g. additional schedules) $\eta_k$ (default 1),
- First moment EMA parameter $\beta_1$ (default 0.9)
- Second moment EMA parameter $\beta_2$ (default 0.999)
- Small positive constant to prevent division with zero $\epsilon$ (default $10^{-8}$)
- Total number of iterations $K$

2: Initialize $s_0 = 0$, $m_0 = 0$, $v_0 = 0, r_0 = 0$

    **for** $k = 0$ **to** $K$ **do**

    Draw mini-batch $i_k$ with associated objective $f^{(i_k)}$

    $g_k \in \partial f^{(i_k)}(w_k)$

    $m_{k+1} = \beta_1 m_k + (1 - \beta_1) d_k \eta_k g_k$

    $v_{k+1} = \beta_2 v_k + (1 - \beta_2) g_k^2$

    $A_{k+1} = \text{diag}(\sqrt{v_{k+1}} + \epsilon)$

    $w_{k+1} = w_k - A_{k+1}^{-1} m_{k+1}$

    $s_{k+1} = \sqrt{\beta_2} s_k + (1 - \sqrt{\beta_2}) d_k \eta_k g_k$

    $r_{k+1} = \sqrt{\beta_2} r_k + (1 - \sqrt{\beta_2}) d_k \eta_k \langle g_k, s_k \rangle_{A_{k+1}^{-1}}$

    $\hat{d}_{k+1} = \frac{r_{k+1}}{(1 - \sqrt{\beta_2}) \|s_{k+1}\|_1}$

    $d_{k+1} = \max\{d_k, \hat{d}_{k+1}\}$

3: **Outputs** Last iterate $w_K$

---

---

**Algorithm 2:** PRODIGY algorithm

---

**Inputs:**
- Initial parameters $w_0$,
- Initial estimate of the distance $d_0$ (default $10^{-6}$),
- Optional sequence of learning rate (to include e.g. additional schedules) $\eta_k$ (default 1),
- First moment EMA parameter $\beta_1$ (default 0.9)
- Second moment EMA parameter $\beta_2$ (default 0.999)
- Small positive constant to prevent division with zero $\epsilon$ (default $10^{-8}$)
- Total number of iterations $K$

Initialize $r_0 = 0$, $s_0 = 0$, $m_0 = 0$, $v_0 = 0$

    **for** $k = 0$ **to** $K$ **do**

    Draw mini-batch $i_k$ with associated objective $f^{(i_k)}$

    $g_k \in \partial f^{(i_k)}(w_k)$

    $m_{k+1} = \beta_1 m_k + (1 - \beta_1) d_k g_k$

    $v_{k+1} = \beta_2 v_k + (1 - \beta_2) d_k^2 g_k^2$

    $r_{k+1} = \sqrt{\beta_2} r_k + (1 - \sqrt{\beta_2}) \eta_k d_k^2 \langle g_k, w_0 - w_k \rangle$

    $s_{k+1} = \sqrt{\beta_2} s_k + (1 - \sqrt{\beta_2}) \eta_k d_k^2 g_k$

    $\hat{d}_{k+1} = \frac{r_{k+1}}{\|s_{k+1}\|_1}$

    $d_{k+1} = \max(d_k, \hat{d}_{k+1})$

    $w_{k+1} = w_k - \eta_k d_k m_{k+1} / (\sqrt{v_{k+1}} + d_k \epsilon)$

    **Outputs** Last iterate $w_K$

---

---

**Algorithm 3:** MECHANIC: A Learning Rate Tuner.

---

1   **Default settings:** $n = 6$, $\beta = (0.9, 0.99, 0.999, 0.9999, 0.99999, 0.999999)$, $\lambda = 0.01$, $s_{\text{init}} = 10^{-8}$.

    **Input:** Base algorithm base, $w_1^{base} \in \mathbb{R}^d$, $\beta \in [0, 1)^n$, $\lambda \in \mathbb{R}$, $s_{\text{init}} \in \mathbb{R}$, $\epsilon = 10^{-8}$

2   **Init:** $v_0 = 0 \in \mathbb{R}^n$, $r_0 = 0 \in \mathbb{R}^n$, $m_0 = 0 \in \mathbb{R}^n$, $w_{\text{ref}} = w_1^{base}$, $\Delta_1 = 0 \in \mathbb{R}^d$, $s_1 = 0 \in \mathbb{R}^n$

3   **for** $k = 1, 2, \ldots$ **do**

4       $g_k \leftarrow \nabla f(w_k, z_k)$

5       Send $g_k$ to base, receive update $u_k$.

6       $\Delta_{k+1} \leftarrow \Delta_k + u_k$

7       $h_k = \left\langle g_k + \dfrac{\lambda(\sum_{i=1}^n s_{t,i})\|g_k\|w_k}{\|w_k\|}, \Delta_k \right\rangle$ ;         `// Note use of` $\Delta_k$ `rather than` $\Delta_{t+1}$`.`

8       $m_k = \max(\beta m_{t-1}, h_k)$

9       $v_k \leftarrow \beta^2 v_{t-1} + h_k^2$

10      $r_k \leftarrow \beta r_{t-1} - s_{t-1} h_k$

11      $r_k \leftarrow \max(0, r_k)$

12      $W_k \leftarrow s_{\text{init}} \cdot m_k + r_k$

13      $s_{t+1} \leftarrow \sqrt{\dfrac{W_t}{v_t + \epsilon}}$

14      $w_{k+1} \leftarrow w_{\text{ref}} + \left(\sum_{i=1}^n s_{t+1,i}\right) \cdot \Delta_{t+1}$

---

**Algorithm 4:** MoMo: Model-based Momentum method.

---

1   **Hyperparameters:** momentum parameter $\beta \in [0, 1)$, maximum learning rate scheduler $\alpha_k > 0$, objective lower bound approximations $f_\star^k$.

2   **Default settings:** $\alpha_k = 1$, $\beta = 0.9$, $f_\star^k = 0$.

3   **Init:** $\bar{f}_0 = f(w_1, s_1)$, $d_0 = \nabla f(w_1, s_1)$, $\tau_0 = \langle d_0, w^1 \rangle$

    **Input:** Initial iterate $w^1 \in \mathbb{R}^d$, $f_\star^k \subset \mathbb{R}$

4   **for** $k = 1$ **to** $K - 1$ **do**

5       $\bar{f}_k = (1 - \beta)f(w_k, s_k) + \beta \bar{f}_{k-1}$

6       $\tau_k = (1 - \beta)\langle \nabla f(w_k, s_k), w_k \rangle + \beta \tau_{k-1}$

7       $d_k = (1 - \beta)\nabla f(w_k, s_k) + \beta d_{k-1}$

8       $h_k = \bar{f}_k + \langle d_k, w_k \rangle - \tau_k$

9       $w_{k+1} = w_k - \min\left\{\alpha_k, \dfrac{(h_k - f_\star^k)_+}{\|d_k\|^2}\right\} d_k$

---

