# OpenReview forum: "How far away are truly hyperparameter-free learning algorithms?"
_TMLR — Accepted by TMLR_

### Review · Reviewer_PfR8 · 2025-02-19

**Summary Of Contributions:**

This is a pure empirical paper that studies how well existing hyperparameter-free methods can generalize across workloads without workload-specific tuning. To answer this question, this work tested several learning-rate-free optimization methods on the AlgoPerf benchmark. The results show that (a) The tested learning-rate-free methods using default hyperparameters from the paper perform worse than the adaptive optimization baselines (i.e., AdamW and NadamW); (b) The performances of some tested learning-rate-free algorithms can improve by tuning the regularization parameters; and (c) Even the best learning-rate-free methods are not clearly better than the AdamW and NadamW baselines.

**Audience:**

Yes

**Broader Impact Concerns:**

There is no ethical concern.

**Claims And Evidence:**

Yes

**Requested Changes:**

1. The writing of this paper needs to be polished.

* There are some confusing and contradictory claims. For example, on page 2, it is claimed that there are two routes to reducing the hyperparameter search, "more automatic tuning method" and "eliminating hyperparameters", and this paper is concerned with the latter one. However, at the end of the same paragraph, it is claimed that "discovering which hyperparameters are most essential to tune can lead to more efficient and robust methods". Also, exploring more robust tuning methods seems to be the topic throughout this paper instead of "eliminating hyperparameters". The difference between these two routes is not clear to me.

* “Bettting”

* In Section 3.3, the meaning of $r(\tau)$ is not clear. Why is $r$ a function of $\tau$?

2. As a research paper, it's better if the author(s) can provide some in-depth analyses about the insights of the (negative) results or inspiring suggestions on improving the hyperparameter-free methods. Otherwise, one may not believe that "the future is bright" as claimed in the conclusion from the current results shown in the paper.

3. The paper analyzing the AlgoPerf competition results [1] is highly related to this work. It presents some hyperparameter-free methods (e.g., schedule free AdamW) that perform much better than those tested in this work. A fair comparison and discussion could improve the significance of this paper.

4. In Tables 6&9, it seems that all the scores are lower than the baseline scores of the AlgoPerf (see Table 1 of [1]). What's the reason?

[1] Accelerating neural network training: An analysis of the AlgoPerf competition. ICLR 2025

**Strengths And Weaknesses:**

**Strengths:**

* This work has proposed an important and interesting question. The generalization abilities of the existing hyperparameter-free methods have not been studied comprehensively.

* The paper is well-organized and easy to follow.

**Weakness:**

* The technical novelty of this paper is poor. All the empirical studies of this work rely on the AlgoPerf benchmark, including the workloads, tuning method, performance metrics, etc. There is no new hyperparameter-free method, evaluation metric, or benchmark workload proposed. The technical difficulty of evaluating the existing learning-rate-free methods using the AlgoPerf benchmark is not clear.

* The significance of the results is limited. I acknowledge this paper's effort to conduct comprehensive tests of the learning-rate-free methods. However, the results seem to be negative and this work didn't provide any insights into the reason for the poor results or any suggestions to improve the existing algorithms.

---

> ### Author Response · Authors · 2025-03-06
> **Response to reviewer**
>
> Dear Reviewer PfR8,
>
> Thank you for your review. We appreciate that you feel our work proposes an important question.
>
> We will first address your general weaknesses, and then answer specific criticisms.
>
> _The technical novelty of this paper is poor… There is no new hyperparameter-free method, evaluation metric, or benchmark workload proposed._
>
> The novel ideas we are bringing to the community with this work are primarily methodological: we believe that the research area of self-tuning/learning-rate-free methods can be much improved using standardized benchmarking tools. This is why we based the evaluation on the AlgoPerf benchmark; it already has buy-in from the community as a useful evaluation tool.
>
> Regarding the significance of the results: in this paper we focused on quantifying performance rather than attempting to explain the detailed failure modes of all the algorithms we studied. The learning trajectories in Appendix B give some hints; some algorithms led to unstable training on workloads, others lead to slow changes in the loss. We agree that providing explanations for algorithm failures would be useful but we believe it is beyond the scope of this work — explaining failures in a causal way can be quite subtle as it often involves feedback between optimizer dynamics and loss landscape properties.
>
> To summarize, our main actionable findings are:
> - Measuring the ability of learning-rate-free methods to generalize across workloads is a useful metric of their effectiveness.
> - Tuning remaining hyperparameters against a robust benchmark gives good evidence based defaults.
> - Methods using mechanisms like Mechanic and Prodigy seem to be the most promising for removing base learning rate.
>
> We believe these findings are novel and will be useful to the optimizer design community. We submitted this work to TMLR specifically because of the acceptance criterion which emphasizes community interest and systematic validation of the effectiveness of methods, and de-emphasizes technical novelty, e.g. “novelty of the studied method[s] is not a necessary criteria for acceptance”.
>
> Regarding the requested changes:
>
> We appreciate this point about automatic tuning vs. eliminating hyperparameters vs. robust tuning methods. To us, hyperparameter elimination is really about eliminating tuning toil. This can be accomplished by an algorithmic change which updates the hyperparameter in real time using data from the current training run (automatic tuning), finding a constant value which works in all practical settings, or some larger algorithmic change which removes the need for any such value in a training algorithm. The end effect of any of these is to eliminate both computer and human time needed to correctly set these hyperparameters. We have edited the introduction to clarify our position here.
>
> With regards to the other typos: thanks for catching them. Indeed the definition of r should have no argument.
>
> _it's better if the author(s) can provide some in-depth analyses about the insights of the (negative) results or inspiring suggestions on improving the hyperparameter-free methods._
>
> We agree that insights on the specific failure modes would be valuable; however these are beyond the scope of our current work. Though we don’t propose any algorithmic improvements ourselves, we believe that highlighting Prodigy and Mechanic as the most promising approaches studied is already useful for the community, as is providing a route to validating new ideas.
>
> Regarding the referenced paper [1]: it is indeed relevant and was released concurrent to our submission. We initially did not focus on schedule-free since that method does not eliminate the base learning rate like the methods we studied here. We have included a comparison of schedule-free to the best methods we found in Appendix F; note that the comparison is approximate due to the hardware differences between the competition and our own study. Nevertheless, schedule-free is better than both prodigy and mechanic in this setting. We added details about the comparison in the discussion.
>
> _In Tables 6&9, it seems that all the scores are lower than the baseline scores of the AlgoPerf (see Table 1 of [1]). What's the reason?_
>
> The benchmark score is not an absolute measurement, but rather depends on the total pool of competitors. The raw times to target would in principle allow for comparisons across different "competitions"; however, we trained on different hardware compared to [1] so any comparisons would be estimates at best.
>
> We noticed that you checked “No” for claims and evidence; beyond the specific criticisms, were there any other claims which were not supported by our work? We believe all of our claims are backed up by adequate evidence, but are happy to discuss further.
>
> We have uploaded an updated manuscript. Let us know if we can answer any more questions, we look forward to hearing your responses.

---

### Review · Reviewer_ies4 · 2025-02-25

**Summary Of Contributions:**

This paper mainly provides an empirical investigation that reviews the previous learning-rate-free methods and tests their performances on the AlgoPerf benchmark with other hyperparameters fixed to default values. It also explores potential improvements of those learning-rate-free methods by calibrating their configurations per workload.

Extensive experimental result shows that:

- There is still a large space for improvement when comparing previous learning-rate-free methods with AdamW/NAdamW baselines
- When carefully calibrated, some learning-rate-free methods can be greatly improved
- The best learning-rate-free method after careful AlgoPerf calibration cannot outperform AdamW/NAdamW

The paper has considered many learning-rate-free methods, including DoG, DoWG, PRODIGY, CoCoB, Mechanic, and MoMo. Among those, the authors have shown that only PRODIGY and Mechanic have the potential to be comparable to baseline methods.

**Audience:**

Yes

**Broader Impact Concerns:**

This work mainly aims to advance the field of machine learning, which has no obvious concerns and thus does not necessarily need a Broader Impact Statement.

**Claims And Evidence:**

Yes

**Requested Changes:**

- If there are some insights on how far the current methods' (e.g., PRODIGY) scheduled or auto-determined learning rates are from the baselines (Adam) and possibly the optimal, it would strengthen the work and may positively influence my opinion on recommendation for acceptance.

**Strengths And Weaknesses:**

## Strength
- The motivation is clear and the presentation is good in terms of the flow and details.
- Detailed explanations of experimental settings (e.g., why training horizons are set) are included.
- Experiments are extensive and show convincing evidence that we are still far away from truly hyperparameter-free learning algorithms.

## Weakness
- Limited novelty and insights. Only the performances are compared, and no in-depth ablations are presented to show further insights.
- Notations are not carefully introduced, e.g., what are $\alpha$ and $t_{\text{hor}}$?

---

> ### Author Response · Authors · 2025-03-06
> **Response to reviewer**
>
> We thank the reviewer for their time, and answer some questions below.
>
> *Notations are not carefully introduced, e.g., what are α and  $t_{hor}$?*
>
> α and  t_hor are defined in Section 3.1. α is the fraction of the maximum allowable training steps (t_max) that is actually used for training; t_hor is the final training horizon (α t_max). t_max is a value provided by the benchmark, and t_hor is the number of steps actually used for training. Setting t_hor < t_max is often valuable for hitting targets quickly, since much of the progress on validation metrics is made during the decay phase of the learning rate schedule.
>
> Regarding insights on methods:
>
> Our work focused on developing a good methodology to fairly benchmark learning rate free algorithms in order to determine which approaches are actually promising; as a result, we did not attempt to diagnose specific failure modes of algorithms, leaving that to future work. We note that effective learning rates can’t be extracted from all the algorithms, as some will change the direction of updates as well.

---

### Review · Reviewer_9q3A · 2025-03-06

**Summary Of Contributions:**

This submission presents a valuable empirical investigation into the potential of learning-rate-free optimization methods as a step toward truly hyperparameter-free training algorithms. The authors have undertaken a significant amount of experimentation, employing the AlgoPerf benchmark and a novel "evidence-based calibration" procedure to evaluate and compare several state-of-the-art algorithms. The results provide important insights into the strengths and limitations of current approaches, with practical implications for both algorithm design and application.

**Audience:**

Yes

**Broader Impact Concerns:**

N.A.

**Claims And Evidence:**

Yes

**Requested Changes:**

I recommend that the authors address the points raised above in a revised version of the manuscript. Specifically, I encourage them to:
- Explore techniques for reducing the dependence on the relative learning rate schedules within the learning-rate-free optimizers.
- Conduct a more detailed analysis of the factors contributing to the observed tuning inefficiency of the learning-rate-free methods.
- Provide a more thorough discussion of the generalizability of the results to other types of workloads and datasets.
- Incorporate statistical significance tests to support the reported performance comparisons.

**Strengths And Weaknesses:**

## Strengths:
- The paper provides a thorough and well-executed empirical study of learning-rate-free methods across a diverse set of workloads. The use of the AlgoPerf benchmark ensures a consistent and relevant evaluation framework. The authors' effort to calibrate these methods using evidence-based defaults is a particularly strong aspect of the work, highlighting the importance of well-justified hyperparameter settings.
- The paper is well-written and organized, making it accessible to readers familiar with the field. The inclusion of supplementary materials, such as detailed training curves and hyperparameter settings, enhances the reproducibility and transparency of the research.
- The findings of this study have direct relevance to practitioners seeking to simplify the hyperparameter tuning process for deep learning models. The paper's emphasis on the importance of well-calibrated defaults and rigorous benchmarking is particularly valuable.

## Weaknesses
- While the empirical evaluation is a significant contribution, the paper lacks substantial algorithmic novelty. The focus is primarily on evaluating existing methods rather than introducing fundamentally new optimization techniques. The “evidence-based defaults” approach is, however, a valuable contribution.
- The sensitivity of the tested algorithms to the training horizon and the reliance on a relative learning rate schedule (warmup/decay) represent a key limitation. While acknowledged by the authors, this dependence undermines the objective of achieving a truly "learning-rate-free" solution. Further investigation into mitigating this dependence within the learning-rate-free optimizers themselves is warranted. Perhaps a dynamic horizon determination based on validation loss would improve the results.
- The observation that learning-rate-free methods did not demonstrate a discernible advantage over AdamW/NadamW, even with comparable tuning budgets, raises concerns. A deeper analysis of the factors contributing to this lack of efficiency is necessary. The authors should explore whether the hyperparameter search space was adequately tailored to the learning-rate-free methods and whether potential interactions between the eliminated learning rate and remaining hyperparameters were properly addressed.
- While AlgoPerf provides a valuable benchmark, it is important to acknowledge its limitations. The authors should provide a more explicit discussion of the potential generalizability (or lack thereof) of the results to other types of workloads, datasets, and model architectures. The absence of model-scaling experiments is a notable omission.
- To strengthen the robustness of the findings, the authors should consider incorporating statistical significance tests to support their comparisons between algorithms.

---

> ### Author Response · Authors · 2025-03-06
> **Response to reviewer**
>
> Dear Reviewer 9q3A,
>
> Thank you for your thoughtful comments. We appreciate your recognition of our efforts to work in a consistent and rigorous evaluation framework. We address some of the concerns and requested changes here.
>
> _Explore techniques for reducing the dependence on the relative learning rate schedules within the learning-rate-free optimizers._
>
> We believe that this is an important area of research for the community, and have highlighted a new, promising approach in Appendix F. Taking a deeper dive here is beyond the scope of this work; we hope to explore these directions more in future work.
>
> _Conduct a more detailed analysis of the factors contributing to the observed tuning inefficiency of the learning-rate-free methods._
>
> We believe that our tuning procedure was reasonably rooted in established methodology, and additionally reflects approaches practitioners might take in analyzing their own problems. In our view this approach allowed for reasonable comparisons between algorithms with quite different properties.
>
> We encourage researchers to include information about non-standard tuning approaches that are useful for their specific algorithms. However conducting a broad “meta-tuning” search can quickly become complicated and is not the subject of this work.
>
> _Provide a more thorough discussion of the generalizability of the results to other types of workloads and datasets._
>
> We are working to improve this in our text, thanks for the suggestion.
>
> _Incorporate statistical significance tests to support the reported performance comparisons._
>
> Each training run (algorithm, workload, tuning horizon, and hyperparameter specification) was evaluated over 5 studies where the median runtime was used to calculate the AlgoPerf scores. Did you have suggestions for additional statistical tests? Given that the algoperf score is a non-linear composite of multiple measurements, is there some sort of bootstrapping approach that would be appropriate here?

---

### Author Response · Authors · 2025-03-14
**Any further clarifications?**

We have uploaded a draft addressing some specific requests and fixing issues noticed by the reviewers. Reviewers: please let us know if there is anything further we can clarify for you!